# Towards a Zero-One Law for Column Subset Selection

**Zhao Song**[*]
University of Washington
magic.linuxkde@gmail.com

**David P. Woodruff**[*]
Carnegie Mellon University
dwoodruf@cs.cmu.edu

**Peilin Zhong**[*]
Columbia University
pz2225@columbia.edu

## Abstract

There are a number of approximation algorithms for NP-hard versions of low rank approximation, such as finding a rank-$k$ matrix $B$ minimizing the sum of absolute values of differences to a given $n$-by-$n$ matrix $A$, $\min_{\text{rank-}k\ B} \|A - B\|_1$, or more generally finding a rank-$k$ matrix $B$ which minimizes the sum of $p$-th powers of absolute values of differences, $\min_{\text{rank-}k\ B} \|A - B\|_p^p$. Many of these algorithms are linear time columns subset selection algorithms, returning a subset of $\text{poly}(k \log n)$ columns whose cost is no more than a $\text{poly}(k)$ factor larger than the cost of the best rank-$k$ matrix. The above error measures are special cases of the following general entrywise low rank approximation problem: given an arbitrary function $g : \mathbb{R} \to \mathbb{R}_{\geq 0}$, find a rank-$k$ matrix $B$ which minimizes $\|A - B\|_g = \sum_{i,j} g(A_{i,j} - B_{i,j})$. A natural question is which functions $g$ admit efficient approximation algorithms? Indeed, this is a central question of recent work studying generalized low rank models. In this work we give approximation algorithms for *every* function $g$ which is approximately monotone and satisfies an approximate triangle inequality, and we show both of these conditions are necessary. Further, our algorithm is efficient if the function $g$ admits an efficient approximate regression algorithm. Our approximation algorithms handle functions which are not even scale-invariant, such as the Huber loss function, which we show have very different structural properties than $\ell_p$-norms, e.g., one can show the lack of scale-invariance causes any column subset selection algorithm to provably require a $\sqrt{\log n}$ factor larger number of columns than $\ell_p$-norms; nevertheless we design the first efficient column subset selection algorithms for such error measures.

## 1 Introduction

A well-studied problem in machine learning and numerical linear algebra, with applications to recommendation systems, text mining, and computer vision, is that of computing a low-rank approximation of a matrix. Such approximations reveal low-dimensional structure, provide a compact way of storing a matrix, and can quickly be applied to a vector.

A commonly used version of the problem is to compute a near optimal low-rank approximation with respect to the Frobenius norm. That is, given an $n \times n$ input matrix $A$ and an accuracy parameter $\epsilon > 0$, output a rank-$k$ matrix $B$ with large probability so that $\|A - B\|_F^2 \leq (1 + \epsilon)\|A - A_k\|_F^2$, where for a matrix $C$, $\|C\|_F^2 = \sum_{i,j} C_{i,j}^2$ is its squared Frobenius norm, and $A_k = \text{argmin}_{\text{rank-}k\ B} \|A - B\|_F$. $A_k$ can be computed exactly using the singular value decomposition (SVD), but takes $O(n^3)$ time in practice and $n^\omega$ time in theory, where $\omega \approx 2.373$ is the exponent of matrix multiplication [1, 2, 3, 4].

Sárlos [5] showed how to achieve the above guarantee with constant probability in $\widetilde{O}(\text{nnz}(A) \cdot k/\epsilon) + n \cdot \text{poly}(k/\epsilon)$ time, where $\text{nnz}(A)$ denotes the number of non-zero entries of $A$. This was improved

---

[*]equal contribution.

in [6, 7, 8, 9, 10] using sparse random projections in $O(\text{nnz}(A)) + n \cdot \text{poly}(k/\epsilon)$ time. Large sparse datasets in recommendation systems are common, such as the Bookcrossing ($100K \times 300K$ with $10^6$ observations) [11] and Yelp datasets ($40K \times 10K$ with $10^5$ observations) [12], and this is a substantial improvement over the SVD.

**Robust Low Rank Approximation.** To understand the role of the Frobenius norm in the algorithms above, we recall a standard motivation for this error measure. Suppose one has $n$ data points in a $k$-dimensional subspace of $\mathbb{R}^d$, where $k \ll d$. We can write these points as the rows of an $n \times d$ matrix $A^*$ which has rank $k$. The matrix $A^*$ is often called the *ground truth matrix*. In a number of settings, due to measurement noise or other kinds of noise, we only observe the matrix $A = A^* + \Delta$, where each entry of the *noise matrix* $\Delta \in \mathbb{R}^{n \times n}$ is an i.i.d. random variable from a certain mean-zero noise distribution $\mathcal{D}$. One method for approximately recovering $A^*$ from $A$ is maximum likelihood estimation. Here one tries to find a matrix $B$ maximizing the log-likelihood: $\max_{\text{rank-}k \ B} \sum_{i,j} \log p(A_{i,j} - B_{i,j})$, where $p(\cdot)$ is the probability density function of the underlying noise distribution $\mathcal{D}$. For example, when the noise distribution is Gaussian with mean zero and variance $\sigma^2$, denoted by $N(0, \sigma^2)$, then the optimization problem is $\max_{\text{rank-}k \ B} \sum_{i,j} \left( \log(1/\sqrt{2\pi\sigma^2}) - (A_{i,j} - B_{i,j})^2/(2\sigma^2) \right)$, which is equivalent to solving the Frobenius norm loss low rank approximation problem defined above.

The Frobenius norm loss, while having nice statistical properties for Gaussian noise, is well-known to be sensitive to outliers. Applying the same maximum likelihood framework above to other kinds of noise distributions results in minimizing other kinds of loss functions. In general, if the density function of the underlying noise $\mathcal{D}$ is $p(z) = c \cdot e^{-g(z)}$, where $c$ is a normalization constant, then the maximum likelihood estimation problem for this noise distribution becomes the following generalized entry-wise loss low rank approximation problem: $\min_{\text{rank-}k \ B} \sum_{i,j} g(A_{i,j} - B_{i,j}) = \min_{\text{rank-}k \ B} \|A - B\|_g$, which is a central topic of recent work on *generalized low-rank models* [13]. For example, when the noise is Laplacian, the entrywise $\ell_1$ loss is the maximum likelihood estimation, which is also robust to sparse outliers. A natural setting is when the noise is a mixture of small Gaussian noise and sparse outliers; this noise distribution is referred to as the *Huber density*. In this case the Huber loss function gives the maximum likelihood estimate [13], where the Huber function [14] is defined to be: $g(x) = x^2/(2\tau)$ if $|x| < \tau/2$, and $g(x) = |x| - \tau/2$ if $|x| \geq \tau$. Another nice property of the Huber error measure is that it is differentiable everywhere, unlike the $\ell_1$-norm, yet still enjoys the robustness properties as one moves away from the origin, making it less sensitive to outliers than the $\ell_2$-norm. There are many other kinds of loss functions, known as $M$-estimators [15], which are widely used as loss functions in robust statistics [16].

Although several specific cases have been studied, such as entry-wise $\ell_p$ loss [17, 18, 19, 20, 21], weighted entry-wise $\ell_2$ loss [22], and cascaded $\ell_p(\ell_2)$ loss [23, 24], the landscape of general entry-wise loss functions remains elusive. There are no results known for any loss function which is not scale-invariant, much less any kind of characterization of which loss functions admit efficient algorithms. This is despite the importance of these loss functions; we refer the reader to [13] for a survey of generalized low rank models. This motivates the main question in our work:

**Question 1.1** (General Loss Functions). *For a given approximation factor $\alpha > 1$, which functions $g$ allow for efficient low-rank approximation algorithms? Formally, given an $n \times d$ matrix $A$, can we find a $\text{rank-}k$ matrix $B$ for which $\|A - B\|_g \leq \alpha \min_{\text{rank-}k \ B'} \|A - B'\|_g$, where for a matrix $C$, $\|C\|_g = \sum_{i \in [n], j \in [d]} g(C_{i,j})$? What if we also allow $B$ to have rank $\text{poly}(k \log n)$?*

For Question 1.1, one has $g(x) = |x|^p$ for $p$-norms, and note the Huber loss function also fits into this framework. Allowing $B$ to have slightly larger rank than $k$, namely, $\text{poly}(k \log n)$, is often sufficient for applications as it still allows for the space savings and computational gains outlined above. These are referred to as bicriteria approximations and are the focus of our work.

**Notation.** Before we present our results, let us briefly introduce the notation. For $n \in \mathbb{Z}_{\geq 0}$, let $[n]$ denote the set $\{1, 2, \cdots, n\}$. Let $A \in \mathbb{R}^{n \times m}$. $A_i$ and $A^j$ denote the $i^{\text{th}}$ column and the $j^{\text{th}}$ row of $A$ respectively. Let $P \subseteq [m], Q \subseteq [n]$. $A_P$ denotes the matrix which is composed by the columns of $A$ with column indices in $P$. Similarly, $A^Q$ denotes the matrix composed by the rows of $A$ with row indices in $Q$. Let $S$ be a set and $s \in \mathbb{Z}_{\geq 0}$. We use $\binom{S}{s}$ to denote the set of all the size-$s$ subsets of $S$.

Table 1: Example functions satisfying both structural properties.

| | $g(x)$ | $\text{ati}_{g,t}$ | $\text{mon}_g$ |
|---|---|---|---|
| HUBER | $\begin{cases} x^2/2 & \|x\| \le \tau \\ \tau(\|x\| - \tau/2) & \|x\| > \tau \end{cases}$ | $O(t)$ | $1$ |
| $\ell_p$ $(p \ge 1)$ | $\|x\|^p/p$ | $O(t^{p-1})$ | $1$ |
| $\ell_1 - \ell_2$ | $2(\sqrt{1 + x^2/2} - 1)$ | $O(t)$ | $1$ |
| GEMAN-MCCLURE | $x^2/(2 + 2x^2)$ | $O(t)$ | $1$ |
| "FAIR" | $\tau^2 \left(\|x\|/\tau - \log(1 + \|x\|/\tau)\right)$ | $O(t)$ | $1$ |
| TUKEY | $\begin{cases} \tau^2/6 \cdot (1 - (1 - (x/\tau)^2)^3) & \|x\| \le \tau \\ \tau^2/6 & \|x\| > \tau \end{cases}$ | $O(t)$ | $1$ |
| CAUCHY | $\tau^2/2 \cdot \log(1 + (x/\tau)^2)$ | $O(t)$ | $1$ |
| QUANTILE $(\tau \in (0,1))$ | $\begin{cases} \tau x & x \ge 0 \\ (\tau - 1)x & x < 0 \end{cases}$ | $1$ | $\max\left(\frac{\tau}{1-\tau}, \frac{1-\tau}{\tau}\right)$ |

## 1.1 Our Results

We studied low rank approximation with respect to general error measures. Our algorithm is a column subset selection algorithm, returning a small subset of columns which span a good low rank approximation. Column subset selection has the benefit of preserving sparsity and interpretability, as described above.

We give a "zero-one law" for such column subset selection problems. We describe two properties on the function $g$ that we need to obtain our low rank approximation algorithms. We also show that if we are missing any one of the properties, then we can find an example function $g$ for which there is no good column subset selection (see Appendix B).

Since we obtain column subset selection algorithms for a wide class of functions, our algorithms must necessarily be bicriteria and have approximation factor at least $\text{poly}(k)$. Indeed, a special case of our class of functions includes entrywise $\ell_1$-low rank approximation, for which it was shown in Theorem G.27 of [18] that any subset of $\text{poly}(k)$ columns incurs an approximation error of at least $k^{\Omega(1)}$. We also show that for the entrywise Huber-low rank approximation, already for $k = 1$, $\sqrt{\log n}$ columns are needed to obtain any constant factor approximation, thus showing that for some of the functions we consider, a dependence on $n$ in our column subset size is necessary.

We note that previously for almost all such functions, it was not known how to obtain any non-trivial approximation factor with any sublinear number of columns.

### 1.1.1 A Zero-One Law

We first state three general properties, the first two of which are structural properties and are necessary and sufficient for obtaining a good approximation from a small subset of columns. The third property is needed for efficient running time.

**Approximate triangle inequality.** For $t \in \mathbb{Z}_{>0}$, we say a function $g(x) : \mathbb{R} \to \mathbb{R}_{\ge 0}$ satisfies the $\text{ati}_{g,t}$-approximate triangle inequality if for any $x_1, x_2, \cdots, x_t \in \mathbb{R}$, $g\left(\sum x_i\right) \le \text{ati}_{g,t} \cdot \sum g(x_i)$.

**Monotone property.** For any parameter $\text{mon}_g \ge 1$, we say function $g(x) : \mathbb{R} \to \mathbb{R}_{\ge 0}$ is $\text{mon}_g$-monotone if for any $x, y \in \mathbb{R}$ with $0 \le |x| \le |y|$, we have $g(x) \le \text{mon}_g \cdot g(y)$.

Many functions including most $M$-estimators [15] and the quantile function [26] satisfy the above two properties. See Table 1 for several examples. We refer the reader to the supplementary, namely Appendix B, for the necessity of these two properties. Our next property is not structural, but rather states that if the loss function has an efficient regression algorithm, then that suffices to efficiently find a small subset of columns spanning a good low rank approximation.

**Regression property.** We say function $g(x) : \mathbb{R} \to \mathbb{R}_{\ge 0}$ has the $(\text{reg}_{g,d}, \mathcal{T}_{\text{reg},g,n,d,m})$-regression property if the following holds: given two matrices $A \in \mathbb{R}^{n \times d}$ and $B \in \mathbb{R}^{n \times m}$, for each $i \in [m]$, let $\text{OPT}_i$ denote $\min_{x \in \mathbb{R}^d} \|Ax - B_i\|_g$. There is an algorithm that runs in $\mathcal{T}_{\text{reg},g,n,d,m}$ time and outputs a matrix $X' \in \mathbb{R}^{d \times m}$ such that $\|AX'_i - B_i\|_g \le \text{reg}_{g,d} \cdot \text{OPT}_i, \forall i \in [m]$ and outputs a vector of

estimated regression costs $v \in \mathbb{R}^d$ such that $\mathrm{OPT}_i \leq v_i \leq \mathrm{reg}_{g,d} \cdot \mathrm{OPT}_i, \forall i \in [m]$. The success probability is at least $1 - 1/\operatorname{poly}(nm)$.

Some functions for which regression itself is non-trivial are e.g., the $\ell_0$-loss function and Tukey function. The $\ell_0$-loss function corresponds to the nearest codeword problem over the reals and has slightly better than an $O(d)$-approximation ([27, 28], see also [20]). For the Tukey function, [29] shows that Tukey regression is NP-hard, and it also gives approximation algorithms. For discussion on regression solvers, we refer the reader to Appendix C.

**Zero-one law (sufficient conditions):** For any function, as long as the above general three properties hold, we can provide an efficient algorithm, as our following main theorem shows.

**Theorem 1.2.** *Given a matrix $A \in \mathbb{R}^{n \times n}$, let $k \geq 1, k' = 2k + 1$. Let $g : \mathbb{R} \to \mathbb{R}_{\geq 0}$ denote a function satisfying the $\mathrm{ati}_{g,k'}$-approximate triangle inequality, the $\mathrm{mon}_g$-monotone property , and the $(\mathrm{reg}_{g,k'}, \mathcal{T}_{\mathrm{reg},g,n,k',n})$-regression property. Let $\mathrm{OPT} = \min_{\mathrm{rank} -k \ A'} \|A' - A\|_g$. There is an algorithm that runs in $\widetilde{O}(n + \mathcal{T}_{\mathrm{reg},g,n,k',n})$ time and outputs a set $S \subseteq [n]$ with $|S| = O(k \log n)$ such that with probability at least $0.99$,*

$$\min_{X \in \mathbb{R}^{|S| \times n}} \|A_S X - A\|_g \leq \mathrm{ati}_{g,k'} \cdot \mathrm{mon}_g \cdot \mathrm{reg}_{g,k'} \cdot O(k \log k) \cdot \mathrm{OPT}.$$

Although the input matrix $A$ in the above statement is a square matrix, it is straightforward to extend the result to the rectangular case. By the above theorem, we can obtain a good subset of columns. To further get a low rank matrix $B$ which is a good low rank approximation to $A$, it is sufficient to take an additional $\mathcal{T}_{\mathrm{reg},g,n,|S|,n}$ time to solve the regression problem.

**Zero-one law (necessary conditions):** In Appendix B.1, we show how to construct a monotone function without approximate triangle inequality such that it is not possible to obtain a good low rank approximation by selecting a small subset of columns.

In Appendix B.2, we discuss a function which has the approximate triangle inequality but is not monotone. We show that for some matrices, there is no small subset of columns which can give a good low rank approximation for such loss function.

### 1.1.2 Lower Bound on the Number of Columns

One may wonder if the $\log n$ blowup in rank is necessary in our theorem. We show some dependence on $n$ is necessary by showing that for the important Huber loss function, at least $\sqrt{\log n}$ columns are required in order to obtain a constant factor approximation for $k = 1$:

**Theorem 1.3.** *Let $H(x)$ denote the following function: $H(x) = \begin{cases} x^2, & \text{if } |x| < 1; \\ |x|, & \text{if } |x| \geq 1. \end{cases}$*

*For any $n \geq 1$, there is a matrix $A \in \mathbb{R}^{n \times n}$ such that, if we select $o(\sqrt{\log n})$ columns to fit the entire matrix, there is no $O(1)$-approximation, i.e., for any subset $S \subseteq [n]$ with $|S| = o(\sqrt{\log n})$,*

$$\min_{X \in \mathbb{R}^{|S| \times n}} \|A_S X - A\|_H \geq \omega(1) \cdot \min_{\mathrm{rank} -1 \ A'} \|A' - A\|_H.$$

Notice that the above function $H(x)$ is always a constant approximation to the Huber function (see Table 1) with $\tau = 1$. Thus, the hardness also holds for the Huber function. For more discussion on our lower bound, we refer the reader to Appendix D.

### 1.2 Overview of our Approach and Related Work

**Low Rank Approximation for General Functions.** A natural approach to low rank approximation is "column subset selection", which has been extensively studied in numerical linear algebra [30, 31, 32, 33, 34, 35, 36, 37, 18, 38]. One can take the column subset selection algorithm for $\ell_p$-low rank approximation in [19] and try to adapt it to general loss functions. Namely, their argument shows that for any matrix $A \in \mathbb{R}^{n \times n}$ there exists a subset $S$ of $k$ columns of $A$, denoted by $A_S \in \mathbb{R}^{n \times k}$, for which there exists a $k \times n$ matrix $V$ for which $\|A_S V - A\|_p^p \leq (k+1)^p \min_{\mathrm{rank-}k \ B'} \|A - B'\|_p^p$; we refer the reader to Theorem 3 of [19]. Given the existence of such a subset $S$, a natural next idea is to then sample a set $T$ of $k$ columns of $A$ uniformly at random. It is then likely the case that if we look

at a random column $A_i$, (1) with probability $1/(k+1)$, $i$ is not among the subset $S$ of $k$ columns out of the $k+1$ columns $T \cup \{i\}$ defining the optimal rank-$k$ approximation to the submatrix $A_{T \cup \{i\}}$, and (2) with probability at least $1/2$, the best rank-$k$ approximation to $A_{T \cup \{i\}}$ has cost at most

$$(2(k+1)/n) \cdot \min_{\text{rank-}k \ B'} \|A - B'\|_p^p. \tag{1}$$

Indeed, (1) follows from $T \cup \{i\}$ being a uniformly random subset of $k+1$ columns, while (2) follows from a Markov bound. The argument in Theorem 7 of [19] is then able to "prune" a $1/(k+1)$ fraction of columns (this can be optimized to a constant fraction) in expectation, by "covering" them with the random set $T$. Recursing on the remaining columns, this procedure stops after $k \log n$ iterations, giving a column subset of size $O(k^2 \log n)$ (which can be optimized to $O(k \log n)$) and an $O(k)$-approximation.

The proof in [19] of the existence of a subset $S$ of $k$ columns of $A$ spanning a $(k+1)$-approximation above is quite general, and one might suspect it generalizes to a large class of error functions. Suppose, for example, that $k = 1$. The idea there is to write $A = A^* + \Delta$, where $A^* = U \cdot V$ is the optimal rank-1 $\ell_p$-low rank approximation to $A$. One then "normalizes" by the error, defining $\widetilde{A}_i^* = A_i^*/\|\Delta_i\|_p$ and letting $s$ be such that $\|\widetilde{A}_s^*\|_p$ is largest. The rank-1 subset $S$ is then just $A_s$. Note that since $\widetilde{A}^*$ has rank-1 and $\|\widetilde{A}_s^*\|_p$ is largest, one can write $\widetilde{A}_j^*$ for every $j \neq s$ as $\alpha_j \cdot \widetilde{A}_s^*$ for $|\alpha_j| \leq 1$. The fact that $|\alpha_j| \leq 1$ is crucial; indeed, consider what happens when we try to "approximate" $A_j$ by $A_s \cdot \frac{\alpha_j \|\Delta_j\|_p}{\|\Delta_s\|_p}$. Then $\|A_j - A_s \alpha_j \|\Delta_j\|_p/\|\Delta_s\|_p\|_p \leq \|A_j - A_j^*\|_p + \|A_j^* - A_s \alpha_j \|\Delta_j\|_p/\|\Delta_s\|_p\|_p = \|\Delta_j\|_p + \|A_j^* - (A_s^* + \Delta_s)\alpha_j \|\Delta_j\|_p/\|\Delta_s\|_p\|_p = \|\Delta_j\|_p + \|\Delta_s \alpha_j \|\Delta_j\|_p/\|\Delta_s\|_p\|_p$, and since the $p$-norm is monotonically increasing and $\alpha_j \leq 1$, the latter is at most $\|\Delta_j\|_p + \|\Delta_s \frac{\|\Delta_j\|_p}{\|\Delta_s\|_p}\|_p$. So far, all we have used about the $p$-norm is the monotone increasing property, so one could hope that the argument could be generalized to a much wider class of functions.

However, at this point the proof uses that the $p$-norm has *scale-invariance*, and so $\|\Delta_s \frac{\|\Delta_j\|_p}{\|\Delta_s\|_p}\|_p = \|\Delta_j\|_p \cdot \|\frac{\Delta_s}{\|\Delta_s\|_p}\|_p = \|\Delta_j\|_p$, and it follows that $\|A_j - A_s \frac{\alpha_j \|\Delta_j\|_p}{\|\Delta_s\|_p}\|_p \leq 2\|\Delta_j\|_p$, giving an overall 2-approximation (recall $k = 1$). But what would happen for a general, not necessarily scale-invariant function $g$? We need to bound $\|\Delta_s \frac{\|\Delta_j\|_g}{\|\Delta_s\|_g}\|_g$. If we could bound this by $O(\|\Delta_j\|_g)$, we would obtain the same conclusion as before, up to constant factors. Consider, though, the "reverse Huber function": $g(x) = x^2$ if $x \geq 1$ and $g(x) = |x|$ for $x \leq 1$. Suppose that $\Delta_s$ and $\Delta_j$ were just 1-dimensional vectors, i.e., real numbers, so we need to bound $g(\Delta_s g(\Delta_j)/g(\Delta_s))$ by $O(g(\Delta_j))$. Suppose $\Delta_s = 1$. Then $g(\Delta_s) = 1$ and $g(\Delta_s g(\Delta_j)/g(\Delta_s)) = g(g(\Delta_j))$ and if $\Delta_j = n$, then $g(g(\Delta_j)) = n^4 = g(\Delta_j)^2$, much larger than the $O(g(\Delta_j))$ we were aiming for.

Maybe the analysis can be slightly changed to correct for these normalization issues? This is not the case, as we show that unlike for $\ell_p$-low rank approximation, for the reverse Huber function *there is no subset of 2 columns of A obtaining better than an $n^{1/4}$-approximation factor.* (See Section D.2 for more details). Further, the lack of scale invariance not only breaks the argument in [19], it shows that combinatorially such functions $g$ behave very differently than $\ell_p$-norms. We show more generally there exist functions, in particular the Huber function, for which one needs to choose $\Omega(\sqrt{\log n})$ columns to obtain a constant factor approximation; we describe this more below. Perhaps more surprisingly, we show a subset of $O(\log n)$ columns suffice to obtain a constant factor approximation to the best rank-1 approximation for any function $g(x)$ which is approximately monotone and has the approximate triangle inequality, the latter implying for any constant $C > 0$ and any $x \in \mathbb{R}_{\geq 0}$, $g(Cx) = O(g(x))$. For $k > 1$, these conditions become: (1) $g(x)$ is monotone non-decreasing in $x$, (2) $g(x)$ is within a $\text{poly}(k)$ factor of $g(-x)$, and (3) for any real number $x \in \mathbb{R}^{\geq 0}$, $g(O(kx)) \leq \text{poly}(k) \cdot g(x)$. We show it is possible to obtain an $O(k^2 \log k)$ approximation with $O(k \log n)$ columns. We give the intuition and main lemma statements for our result in Section 2, deferring proofs to the supplementary material.

Even for $\ell_p$-low rank approximation, our algorithms slightly improve and correct a minor error in [19] which claims in Theorem 7 an $O(k)$-approximation with $O(k \log n)$ columns for $\ell_p$-low rank approximation. However, their algorithm actually gives an $O(k \log n)$-approximation with $O(k \log n)$ columns. In [19] it was argued that one expects to pay a cost of $O(k/n) \cdot \min_{\text{rank-}k \ B'} \|A - B'\|_p^p$ per column as in (1), and since each column is only counted in one iteration, summing over the columns gives $O(k) \cdot \min_{\text{rank-}k \ B'} \|A - B'\|_p$ total cost. The issue is that the value of $n$ is changing in each

---

**Algorithm 1** Low rank approximation algorithm for general functions

---

1: **procedure** GENERALFUNCTIONLOWRANKAPPROX($A \in \mathbb{R}^{n \times n}, k \in \mathbb{Z}_{\geq 1}, g : \mathbb{R} \to \mathbb{R}_{\geq 0}$)
2:     Initialization: $T_0 \leftarrow [n], i \leftarrow 1, r \leftarrow 0$
3:     **for** $|T_{i-1}| \geq 1000k$ **do**
4:         **for** $j = 1 \to \log n$ **do**
5:             Sample $S_i^{(j)}$ from $\binom{T_{i-1}}{2k}$ uniformly at random
6:             Solve the $\mathrm{reg}_{g,2k}$-approximate regression $\min_{x \in \mathbb{R}^{2k}} \|A_{S_i^{(j)}} x - A_t\|_g$ for each $t \in$
    $T_{i-1} \setminus S_i^{(j)}$, and let $v_{i,t}^{(j)}$ be the $\mathrm{reg}_{g,2k}$-estimated regression cost        ▷ See Section 1.1.1 for
regression property
7:             $R_i^{(j)} \leftarrow \{t \mid v_{i,t}^{(j)} \text{ is the bottom } |T_{i-1} \setminus S_i^{(j)}|/20 \text{ largest value in } \{v_{i,t'}^{(j)} \mid t' \in T_{i-1} \setminus$
    $S_i^{(j)}\}\}$
8:             $c_i^{(j)} \leftarrow \sum_{t \in R_i^{(j)}} v_{i,t}^{(j)}$
9:         **end for**
10:         $j^* \leftarrow \arg\min_{j \in [\log n]} \left\{ c_i^{(j)} \right\}$
11:         $S_i \leftarrow S_i^{(j^*)}, R_i \leftarrow R_i^{(j^*)}, T_i \leftarrow T_{i-1} \setminus (S_i \cup R_i)$
12:         $r \leftarrow i$
13:         $i \leftarrow i + 1$
14:     **end for**
15:     **return** $S = T_r \cup \bigcup_{i \in [r]} S_i$       ▷ It is easy to see $r \leq O(\log n)$ from the above procedure
16: **end procedure**

---

iteration, so if in the $i$-th iteration it is $n_i$, then we could pay $n_i \cdot O(k/n_i) \cdot \min_{\text{rank-}k \ B'} \|A - B'\|_p = O(k) \cdot \min_{\text{rank-}k \ B'} \|A - B'\|_p$ in each of $O(\log n)$ iterations, giving $O(k \log n)$ approximation ratio. In contrast, our algorithm achieves an $O(k \log k)$ approximation ratio for $\ell_p$-low rank approximation as a special case, which gives the first $O(1)$ approximation in nearly linear time for any constant $k$ for $\ell_p$ norms. Our analysis is finer in that we show not only do we expect to pay a cost of $O(k/n_i) \cdot \min_{\text{rank-}k \ B'} \|A - B'\|_p^p$ per column in iteration $i$, we pay $O(k/n_i)$ times the cost of the best rank-$k$ approximation to $A$ *after the most costly $n/k$ columns have been removed*; thus we pay $O(k/n_i)$ times a *residual cost* with the top $n/k$ columns removed. This ultimately implies any column's cost can contribute in at most $O(\log k)$ of $O(\log n)$ recursive calls, replacing an $O(\log n)$ factor with an $O(\log k)$ factor in the approximation ratio. This also gives the first $\mathrm{poly}(k)$-approximation for $\ell_0$-low rank approximation, studied in [20], improving the $O(k^2 \log(n/k))$-approximation there to $O(k^2 \log k)$ and giving the first constant approximation for constant $k$.

## 2   Algorithm for General Loss Low Rank Approximation

Our algorithm is presented in Algorithm 1. First, let us briefly analyze the running time. Consider fixed $i \in [r], j \in [\log n]$. Sampling $S_i^{(j)}$ takes $O(k)$ time. Solving $\mathrm{reg}_{g,2k}$-approximate regression $\min_x \|A_{S_i^{(j)}} x - A_t\|_g$ for all $t \in T_{i-1} \setminus S_i^{(j)}$ takes $\mathcal{T}_{\mathrm{reg},g,n,2k,|T_{i-1} \setminus S_i^{(j)}|} \leq \mathcal{T}_{\mathrm{reg},g,n,2k+1,n}$ time. Since finding $|T_{i-1} \setminus S_i^{(j)}|/20$ smallest element can be done in $O(n)$ time, $R_i^{(j)}$ can be computed in $O(n)$ time. Thus the inner loop takes $O(n + \mathcal{T}_{\mathrm{reg},g,n,2k+1,n})$ time. Since $r = O(\log n)$, the total running time over all $i, j$ is $O((n + \mathcal{T}_{\mathrm{reg},g,n,2k+1,n}) \log^2 n)$. In the remainder of the section, we will sketch the proof of the correctness. For the missing proofs, we refer the reader to Appendix A.

### 2.1   Properties of Uniform Column Sampling

Let us first introduce some useful notation. Consider a rank-$k$ matrix $M^* \in \mathbb{R}^{n \times m}$. For a set $H \subseteq [m]$, let $R_{M^*}(H) \subseteq H$ be a set such that

$$R_{M^*}(H) = \arg\max_{P : P \subseteq H} \left\{ \left| \det \left( (M^*)_P^Q \right) \right| \ \Big| \ |P| = |Q| = \mathrm{rank}(M_H^*), Q \subseteq [n] \right\}.$$

where $\det(C)$ denotes the determinant of a square matrix $C$. Notice that in the above formula, the maximum is over all possible choices of $P$ and $Q$ while $R_{M^*}(H)$ only takes the value of the corresponding $P$. By Cramer's rule, if we use a linear combination of the columns of $M^*_{R_{M^*}(H)}$ to express any column of $M^*_H$, the absolute value of every fitting coefficient will be at most 1. For example, consider a rank $k$ matrix $M^* \in \mathbb{R}^{n \times (k+1)}$ and $H = [k+1]$. Let $P \subseteq [k+1], Q \subseteq [n], |P| = |Q| = k$ be such that $|\det((M^*)^Q_P)|$ is maximized. Since $M^*$ has rank $k$, we know $\det((M^*)^Q_P) \neq 0$ and thus the columns of $M^*_P$ are independent. Let $i \in [k+1] \setminus P$. Then the linear equation $M^*_P x = M^*_i$ is feasible and there is a unique solution $x$. Furthermore, by Cramer's rule $x_j = \frac{\det((M^*)^Q_{[k+1]\setminus\{j\}})}{\det((M^*)^Q_P)}$. Since $|\det((M^*)^Q_P)| \geq |\det((M^*)^Q_{[k+1]\setminus\{j\}})|$, we have $\|x\|_\infty \leq 1$.

Consider an arbitrary matrix $M \in \mathbb{R}^{n \times m}$. We can write $M = M^* + N$, where $M^* \in \mathbb{R}^{n \times m}$ is an arbitrary rank-$k$ matrix, and $N \in \mathbb{R}^{n \times m}$ is the residual matrix. The following lemma shows that, if we randomly choose a subset $H \subseteq [m]$ of $2k$ columns, and we randomly look at another column $i$, then with constant probability, the absolute values of all the coefficients of using a linear combination of the columns of $M^*_H$ to express $M^*_i$ are at most 1, and furthermore, if we use the same coefficients to use columns of $M_H$ to fit $M_i$, then the fitting cost is proportional to $\|N_H\|_g + \|N_i\|_g$.

**Lemma 2.1.** *Given a matrix* $M \in \mathbb{R}^{n \times m}$ *and a parameter* $k \geq 1$, *let* $M^* \in \mathbb{R}^{n \times m}$ *be an arbitrary rank-k matrix. Let* $N = M - M^*$. *Let* $H \subseteq [m]$ *be a uniformly random subset of* $[m]$, *and let* $i$ *denote a uniformly random index sampled from* $[m] \setminus H$. *Then* (I) $\Pr[i \notin R_{M^*}(H \cup \{i\})] \geq 1/2$; (II) *If* $i \notin R_{M^*}(H \cup \{i\})$, *then there exist* $|H|$ *coefficients* $\alpha_1, \alpha_2, \cdots, \alpha_{|H|}$ *for which* $M^*_i = \sum_{j=1}^{|H|} \alpha_j (M^*_H)_j, \forall j \in [|H|], |\alpha_j| \leq 1$, *and* $\min_{x \in \mathbb{R}^{|H|}} \|M_H x - M_i\|_g \leq \mathrm{ati}_{g,|H|+1} \cdot \mathrm{mon}_g \cdot \left( \|N_i\|_g + \sum_{j=1}^{|H|} \|(N_H)_j\|_g \right)$.

Notice that part (II) of the above lemma does not depend on any randomness of $H$ or $i$. By applying part (I) of the above lemma, it is enough to prove that if we randomly choose a subset $H$ of $2k$ columns, there is a constant fraction of columns that each column $M^*_i$ can be expressed by a linear combination of columns in $M^*_H$, and the absolute values of all the fitting coefficients are at most 1. Because of Cramer's rule, it thus suffices to prove the following lemma.

**Lemma 2.2.**

$$\Pr_{H \sim \binom{[m]}{2k}} \left[ \left| \left\{ i \mid i \in [m] \setminus H, i \notin R_{M^*}(H \cup \{i\}) \right\} \right| \geq (m - 2k)/4 \right] \geq 1/4.$$

## 2.2 Correctness of the Algorithm

We write the input matrix $A$ as $A^* + \Delta$, where $A^* \in \mathbb{R}^{n \times n}$ is the best rank-$k$ approximation to $A$, and $\Delta \in \mathbb{R}^{n \times n}$ is the residual matrix with respect to $A^*$. Then $\|\Delta\|_g = \sum_{i=1}^n \|\Delta_i\|_g$ is the optimal cost. As shown in Algorithm 1, our approach iteratively eliminates all the columns. In each iteration, we sample a subset of columns, and use these columns to fit other columns. We drop a constant fraction of columns which have a good fitting cost. Suppose the indices of the columns surviving after the $i$-th outer iteration are $T_i = \{t_{i,1}, t_{i,2}, \cdots, t_{i,m_i}\} \subseteq [n]$. Without loss of generality, we can assume $\|\Delta_{t_{i,1}}\|_g \geq \|\Delta_{t_{i,2}}\|_g \geq \cdots \geq \|\Delta_{t_{i,m_i}}\|_g$. The following claim shows that if we randomly sample $2k$ column indices $H$ from $T_i$, then the cost of $\Delta_H$ will not be large.

**Claim 2.3.** *If* $|T_i| = m_i \geq 1000k$, $\Pr_{H \sim \binom{T_i}{2k}} \left[ \sum_{j \in H} \|\Delta_j\|_g \leq 400 \frac{k}{m_i} \sum_{j=\frac{m_i}{100k}}^{m_i} \|\Delta_{t_{i,j}}\|_g \right] \geq \frac{19}{20}$.

By an averaging argument, in the following claim, we can show that there is a constant fraction of columns in $T_i$ whose optimal cost is also small.

**Claim 2.4.** *If* $|T_i| = m_i \geq 1000k$, $\left| \left\{ t_{i,j} \mid t_{i,j} \in T_i, \|\Delta_{t_{i,j}}\|_g \geq \frac{20}{m_i} \sum_{j'=\frac{m_i}{100k}}^{m_i} \|\Delta_{t_{i,j'}}\|_g \right\} \right| \leq \frac{1}{5} m_i$.

By combining Lemma 2.2, part (II) of Lemma 2.1 with the above two claims, it is sufficient to prove the following core lemma. It says that if we randomly choose a subset of $2k$ columns from $T_i$, then we can fit a constant fraction of the columns from $T_i$ with a small cost.

**Lemma 2.5.** *If* $|T_i| = m_i \geq 1000k$,

$$\Pr_{H \sim \binom{T_i}{2k}} \left[ \left| \left\{ j \,\middle|\, j \in T_i, \min_{x \in \mathbb{R}^{|H|}} \|A_H x - A_j\|_g \leq C_1 \cdot \frac{1}{m_i} \cdot \sum_{j'=\frac{m_i}{100k}}^{m_i} \|\Delta_{t_{i,j'}}\|_g \right\} \right| \geq \frac{1}{20} m_i \right] \geq \frac{1}{5},$$

*where* $C_1 = 500 \cdot k \cdot \mathrm{ati}_{g,|S|+1} \cdot \mathrm{mon}_g$.

Let us briefly explain why the above lemma is enough to prove the correctness of our algorithm. For each column $j \in [m]$, either the column $j$ is in $T_r$ and is selected by the end of the algorithm, or $\exists i < r$ such that $j \in T_i \setminus T_{i+1}$. If $j \in T_i \setminus T_{i+1}$, then by the above lemma, we can show that with high probability, $\min_x \|A_{S_{i+1}} x - A_j\|_g \leq O(C_1 \|\Delta\|_1 / |T_i|)$. Thus, $\min_X \|A_{S_{i+1}} X - A_{T_i \setminus T_{i+1}}\|_g \leq O(C_1 \|\Delta\|_1)$. It directly gives a $O(rC_1) = O(C_1 \log n)$ approximation. For the detailed proof of Theorem 1.2, we refer the reader to Appendix A.

# 3 Experiments

We show that with the Huber loss low rank approximation, it is possible to outperform the SVD and entrywise $\ell_1$-low rank approximation on certain noise distributions. Even bi-criteria solutions can work very well. This motivates our study of general entry-wise loss functions.

Suppose the noise of the input matrix is a mixture of small Gaussian noise and sparse outliers. Consider an extreme case: the data matrix $A \in \mathbb{R}^{n \times n}$ is a block diagonal matrix which contains three blocks: one block has size $n_1 \times n_1$ ($n_1 = \Theta(n)$) which has uniformly small noise (every entry is $\Theta(1/\sqrt{n})$), another block has only one entry which is a large outlier (with value $\Theta(n^{0.8})$), and the third matrix is the ground truth matrix with size $n_3 \times n_3$ ($n_3 = \Theta(n^{0.6})$) where the absolute value of each entry is at least $1/n^{o(1)}$ and at most $n^{o(1)}$. If we apply Frobenius norm rank-1 approximation, then since $(n^{0.8})^2 > (n^{0.6})^2 \cdot n^{o(1)}$ and $(n^{0.8})^2 > n^2 \cdot (1/\sqrt{n})^2$, we can only learn the large outlier. If we apply entry-wise $\ell_1$ norm rank-1 approximation, then since $n^2 \cdot 1/\sqrt{n} > (n^{0.6})^2 \cdot n^{o(1)}$ and $n^2 \cdot 1/\sqrt{n} > n^{0.8}$, we can only learn the uniformly small noise. But if we apply Huber loss rank-1 approximation, then we can learn the ground truth matrix.

A natural question is: can bi-criteria Huber loss low rank approximation also learn the ground truth matrix under certain noise distributions? We did experiments to answer this question.

**Parameters.** In each iteration, we choose $2k$ columns to fit the remaining columns, and we drop half of the columns with smallest regression cost. In each iteration, we repeat 20 times to find the best $2k$ columns. At the end, if there are at most $4k$ columns remaining, we finish our algorithm. We choose to optimize the Huber loss function, i.e., $f(x) = \frac{1}{2} x^2$ for $x \leq 1$, and $f(x) = |x| - \frac{1}{2}$ for $x > 1$.

**Data.** We evaluate our algorithms on several input data matrix $A \in \mathbb{R}^{n \times n}$ sizes, for $n \in \{200, 300, 400, 500\}$. For rank-1 bi-criteria solutions, the output rank is given in Table 2.

Table 2: The output rank of our algorithm for different input sizes and for $k = 1$.

| $n$ | 200 | 300 | 400 | 500 |
|---|---|---|---|---|
| Output rank | 12 | 12 | 14 | 14 |

$A$ is constructed as a block diagonal matrix with three blocks. The first block has size $\frac{4}{5}n \times \frac{4}{5}n$. It contains many copies of $k'$ different columns where $k'$ is equal to the output rank corresponding to $n$ (see Table 2). The entry of a column is uniformly drawn from $\{-5/\sqrt{n}, 5/\sqrt{n}\}$. The second block is the ground truth matrix. It is generated by $1/\sqrt{k'} \cdot U \cdot V^\top$ where $U, V \in \mathbb{R}^{n \times k'}$ are two i.i.d. random Gaussian matrices. The last block is a size $k' \times k'$ diagonal matrix where each diagonal entry is a sparse outlier with magnitude of absolute value $5 \cdot n^{0.8}$.

**Experimental Results.** We compare our algorithm with Frobenius norm low rank approximation and entry-wise $\ell_1$ loss low rank approximation algorithms [18]. To make it comparable, we set the target rank of previous algorithms to be the output rank of our algorithm. In Figure 1, we can see that the ground truth matrix is well covered by our Huber loss low rank approximation. In Figure 2, we show that our algorithm indeed gives a good solution with respect to the Huber loss.

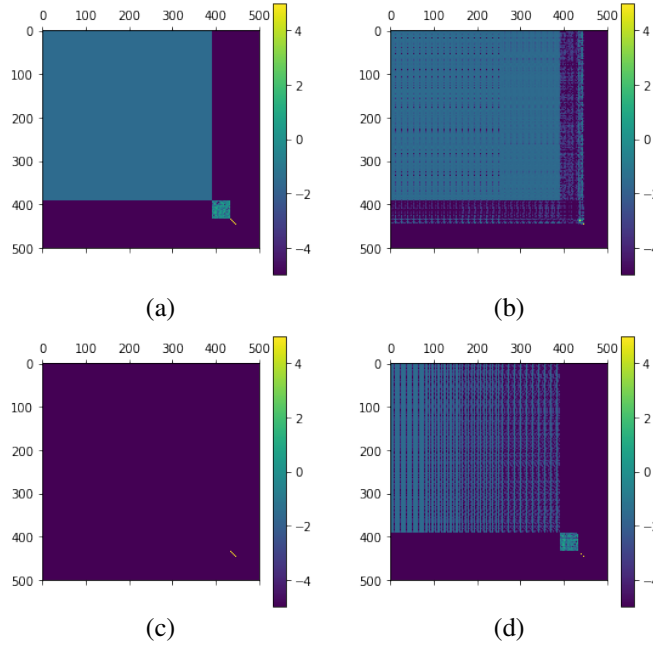

<div align="center">(a)       (b)</div>

<div align="center">(c)       (d)</div>

Figure 1: The input data has size $500 \times 500$. The color indicates the logarithmic magnitude of the absolute value of each entry. (a) is the input matrix. It contains 3 blocks on its diagonal. The top-left one has uniformly small noise. The central one is the ground truth. The bottom-right one contains sparse outliers. Each block has rank 14. So the rank of the input matrix is $3 \times 14 = 42$. (b) is the entry-wise $\ell_1$ loss rank-14 approximation given by [18]. As shown above, it mainly covers the small noise, but loses the information of the ground truth. (c) is the Frobenius norm rank-14 approximation given by the top 14 singular vectors. As shown in the figure, it mainly covers the outliers. However, it loses the information of the ground truth. (d) is the rank-1 bi-criteria solution given by our algorithm. As we can see, it can cover the ground truth matrix quite well.

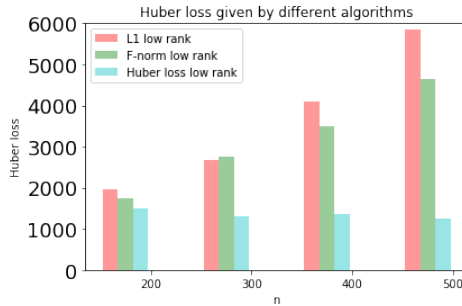

Figure 2: The Huber loss given by different algorithms. The red bar is for the entrywise $\ell_1$ low rank approximation algorithm [18]. The green bar is for traditional PCA. The blue bar is for our algorithm. For input size $n = 200, 300$, all the algorithms output rank-12 approximations. For input size $n = 400, 500$, all the algorithms output rank-14 approximations.

**Acknowledgments.** David P. Woodruff was supported in part by Office of Naval Research (ONR) grant N00014- 18-1-2562. Part of this work was done while he was visiting the Simons Institute for the Theory of Computing. Peilin Zhong was supported in part by NSF grants (CCF-1703925, CCF-1421161, CCF-1714818, CCF-1617955 and CCF-1740833), Simons Foundation (#491119 to Alexandr Andoni), Google Research Award and a Google Ph.D. fellowship. Part of this work was done while Zhao Song and Peilin Zhong were interns at IBM Research - Almaden and while Zhao Song was visiting the Simons Institute for the Theory of Computing.

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
