[Supplementary Material]

# A  Missing Proofs in Section 2

## A.1  Proof of Lemma 2.1

*Proof.* (I) Since $\text{rank}(M^*_{H \cup \{i\}}) \leq \text{rank}(M^*) = k$, $|R_{M^*}(H \cup \{i\})| \leq k$. Note that $H$ is sampled from $\binom{[m]}{2k}$ uniformly at random, $i$ is sampled from $[m] \backslash H$ uniformly at random, and $|H \cup \{i\}| = 2k + 1$. By symmetry we have

$$\Pr_{H \sim \binom{[m]}{2k}, i \sim [m] \backslash H} [i \notin R_{M^*}(H \cup \{i\})] \geq 1 - \frac{k}{2k+1} \geq 1/2.$$

(II) Since $i \notin R_{M^*}(H \cup \{i\})$, by Cramer's rule, there exist $\alpha_1, \alpha_2, \cdots, \alpha_{|H|}$ such that $M^*_i = \sum_{j=1}^{|H|} \alpha_j \cdot (M^*_H)_j$ and $\forall j \in [|H|], |\alpha_j| \leq 1$. Then we have

$$\min_{x \in \mathbb{R}^{|H|}} \|M_H x - M_i\|_g \leq \left\| \sum_{j=1}^{|H|} (M_H)_j \alpha_j - M_i \right\|_g$$

$$= \left\| \sum_{j=1}^{|H|} (M^*_H)_j \alpha_j - M^*_i + \sum_{j=1}^{|H|} (N_H)_j \alpha_j - N_i \right\|_g$$

$$= \left\| \sum_{j=1}^{|H|} (N_H)_j \alpha_j - N_i \right\|_g$$

$$\leq \text{ati}_{g,|H|+1} \cdot \left( \| - N_i \|_g + \sum_{j=1}^{|H|} \|(N_H)_j \alpha_j\|_g \right)$$

$$\leq \text{ati}_{g,|H|+1} \cdot \left( \| - N_i \|_g + \text{mon}_g \cdot \sum_{j=1}^{|H|} \|(N_H)_j\|_g \right)$$

$$\leq \text{ati}_{g,|H|+1} \cdot \left( \text{mon}_g \cdot \|N_i\|_g + \text{mon}_g \cdot \sum_{j=1}^{|H|} \|(N_H)_j\|_g \right)$$

$$\leq \text{ati}_{g,|H|+1} \cdot \text{mon}_g \cdot \left( \|N_i\|_g + \sum_{j=1}^{|H|} \|(N_H)_j\|_g \right),$$

where the second step follows from $M = M^* + N$, the third step follows from $\sum_{j=1}^{|H|} (M^*_H)_j \alpha_j - M^*_i = 0$, the fourth step follows from the approximate triangle inequality, the fifth step follows from the fact that $|\alpha_j| \leq 1$ and $g$ is $\text{mon}_g$-monotone, and the sixth step follows from that $g$ is $\text{mon}_g$-monotone.

$\square$

## A.2  Proof of Lemma 2.2

*Proof.* Using Part (I) of Lemma 2.1, we have

$$\Pr_{H \sim \binom{[m]}{2k}, i \sim [m] \backslash H} [i \notin R_{M^*}(H \cup \{i\})] \geq 1/2.$$

For each set $H$, we define $P_H = \Pr_{i \sim [m] \backslash H}[i \notin R_{M^*}(H \cup \{i\})]$. We have

$$1 \geq \frac{1}{\binom{m}{2k}} \sum_{H \in \binom{[m]}{2k}} P_H \geq 1/2. \tag{2}$$

We can show

$$\frac{1}{\binom{m}{2k}} \left| \left\{ H \ \middle| \ H \in \binom{[m]}{2k}, P_H \geq 1/4 \right\} \right|$$

$$= \frac{1}{\binom{m}{2k}} \sum_{H \in \binom{[m]}{2k}, P_H \geq 1/4} 1$$

$$\geq \frac{1}{\binom{m}{2k}} \sum_{H \in \binom{[m]}{2k}, P_H \geq 1/4} P_H$$

$$\geq \frac{1}{2} - \frac{1}{\binom{m}{2k}} \sum_{H \in \binom{[m]}{2k}, P_H < 1/4} P_H$$

$$\geq \frac{1}{2} - \frac{1}{\binom{m}{2k}} \sum_{H \in \binom{[m]}{2k}, P_H < 1/4} \frac{1}{4}$$

$$\geq \frac{1}{2} - \frac{1}{\binom{m}{2k}} \binom{m}{2k} \frac{1}{4}$$

$$= \frac{1}{4},$$

where the second step follows since $1 \geq P_H$, the third step follows since Eq. (2), the fourth step follows since $P_H < 1/4$.

Thus, we have

$$\left| \left\{ H \ \middle| \ H \in \binom{[m]}{2k}, P_H \geq 1/4 \right\} \right| \geq \binom{m}{2k} / 4.$$

Recall the definition of $P_H$, we have

$$\left| \left\{ H \ \middle| \ H \in \binom{[m]}{2k}, \Pr_{i \sim [m] \setminus H}[i \notin R_{M^*}(H \cup \{i\})] \geq 1/4 \right\} \right| \geq \binom{m}{2k} / 4,$$

which implies

$$\Pr_{H \sim \binom{[m]}{2k}} \left[ \left| \left\{ i \ \middle| \ i \in [m] \setminus H, i \notin R_{M^*}(H \cup \{i\}) \right\} \right| \geq (m - 2k)/4 \right] \geq 1/4.$$

$\square$

### A.3   Proof of Claim 2.3

*Proof.* For simplicity, we omit $i$ in all the subscripts in this proof.

$$\Pr_{H \sim \binom{T}{2k}} \left[ \sum_{j \in H} \|\Delta_j\|_g \leq 400 \frac{k}{m} \sum_{j=\frac{m}{100k}}^{m} \|\Delta_{t_j}\|_g \right]$$

$$= \Pr_{H \sim \binom{T}{2k}} \left[ \sum_{j \in H} \|\Delta_j\|_g \leq 400 \frac{k}{m} \sum_{j=\frac{m}{100k}}^{m} \|\Delta_{t_j}\|_g \ \middle| \ \exists j \leq \frac{m}{100k}, t_j \in H \right] \cdot \Pr_{H \sim \binom{T}{2k}} \left[ \exists j \leq \frac{m}{100k}, t_j \in H \right]$$

$$+ \Pr_{H \sim \binom{T}{2k}} \left[ \sum_{j \in H} \|\Delta_j\|_g \leq 400 \frac{k}{m} \sum_{j=\frac{m}{100k}}^{m} \|\Delta_{t_j}\|_g \ \middle| \ \forall j \leq \frac{m}{100k}, t_j \notin H \right] \cdot \Pr_{H \sim \binom{T}{2k}} \left[ \forall j \leq \frac{m}{100k}, t_j \notin H \right]$$

$$\leq \underbrace{\Pr_{H \sim \binom{T}{2k}} \left[ \exists j \leq \frac{m}{100k}, t_j \in H \right]}_{C_1} + \underbrace{\Pr_{H \sim \binom{T}{2k}} \left[ \sum_{j \in H} \|\Delta_j\|_g \leq 400 \frac{k}{m} \sum_{j=\frac{m}{100k}}^{m} \|\Delta_{t_j}\|_g \ \middle| \ \forall j \leq \frac{m}{100k}, t_j \notin H \right]}_{C_2}$$

It remains to upper bound the terms $C_1$ and $C_2$. We can upper bound $C_1$:

$$
\begin{aligned}
C_1 &= 1 - (1 - \frac{m/100k}{m}) \cdot (1 - \frac{m/100k}{m-1}) \cdot \cdots \cdot (1 - \frac{m/100k}{m-2k+1}) \\
&\leq 1 - (1 - \frac{m/100k}{m/2})^{2k} \\
&\leq 1 - (1 - \frac{1}{25}) \\
&= \frac{1}{25},
\end{aligned}
$$

where the second step follows since $m \geq 1000k$.

Using Markov's inequality,

$$
C_2 \leq \frac{\displaystyle \mathbf{E}_{H \sim \binom{T}{2k}} \left[ \sum_{j \in H} \|\Delta_j\|_g \leq 400 \frac{k}{m} \sum_{j=\frac{m}{100k}}^{m} \|\Delta_{t_j}\|_g \,\middle|\, \forall j \leq \frac{m}{100k}, t_j \notin H \right]}{\displaystyle 400 \frac{k}{m} \sum_{j=\frac{m}{100k}}^{m} \|\Delta_{t_j}\|_g} \leq 1/100,
$$

where the second step follows since

$$
\begin{aligned}
\mathbf{E}_{H \sim \binom{T}{2k}} &\left[ \sum_{j \in H} \|\Delta_j\|_g \leq 400 \frac{k}{m} \sum_{j=\frac{m}{100k}}^{m} \|\Delta_{t_j}\|_g \,\middle|\, \forall j \leq \frac{m}{100k}, t_j \notin H \right] \\
&\leq \frac{2k}{m - m/100k} \sum_{j=\frac{m}{100k}}^{m} \|\Delta_{t_j}\|_g \\
&\leq 4 \frac{k}{m} \sum_{j=\frac{m}{100k}}^{m} \|\Delta_{t_j}\|_g
\end{aligned}
$$

$\square$

## A.4   Proof of Claim 2.4

*Proof.* For simplicity, we omit $i$ in all the subscripts in this proof.

$$
\begin{aligned}
&\left| \left\{ t_j \,\middle|\, t_j \in T, \|\Delta_{t_j}\|_g \geq \frac{20}{m} \sum_{j'=\frac{m}{100k}}^{m} \|\Delta_{t_{j'}}\|_g \right\} \right| \\
&\leq \left| \left\{ t_j \,\middle|\, t_j \in T, j \leq \frac{m}{100k} \right\} \right| + \left| \left\{ t_j \,\middle|\, t_j \in T, j > \frac{m}{100k}, \|\Delta_{t_j}\|_g \geq \frac{20}{m} \sum_{j'=\frac{m}{100k}}^{m} \|\Delta_{t_{j'}}\|_g \right\} \right| \\
&\leq \left| \left\{ t_j \,\middle|\, t_j \in T, j \leq \frac{m}{100k} \right\} \right| + \left| \left\{ t_j \,\middle|\, t_j \in T, j > \frac{m}{100k}, \|\Delta_{t_j}\|_g \geq \frac{10}{m - \frac{m}{100k}} \sum_{j'=\frac{m}{100k}}^{m} \|\Delta_{t_{j'}}\|_g \right\} \right| \\
&\leq \frac{m}{100k} + \frac{1}{10}(m - \frac{m}{100k}) \\
&\leq \frac{1}{5}m,
\end{aligned}
$$

where the second step follows since $\frac{20}{m} \geq \frac{10}{m-m/100k}$ $\square$

## A.5   Proof of Lemma 2.5

*Proof.* For simplicity, we omit $i$ in all the subscirptis in this proof.

Let $M = A_T$, $M^* = A_T^*$ and $N = \Delta_T$. Then we can apply Lemma 2.2 and part (II) of Lemma 2.1:

$$\Pr_{H \sim \binom{T}{2k}} \left[ \left| \left\{ j \in T \;\middle|\; \min_{x \in \mathbb{R}^{|H|}} \|A_H x - A_j\|_g \leq \mathrm{ati}_{g,|H|+1} \cdot \mathrm{mon}_g \cdot \left( \|\Delta_j\|_g + \sum_{j'=1}^{|H|} \|(\Delta_H)_{j'}\|_g \right) \right\} \right| \geq \frac{m}{4} \right] \geq \frac{1}{4} \tag{3}$$

By Claim 2.3, we have

$$\Pr_{H \sim \binom{T}{2k}} \left[ \sum_{j=1}^{|H|} \|(\Delta_H)_j\|_g \leq 400 \frac{k}{m} \sum_{j=\frac{m}{100k}}^{m} \|\Delta_{t_j}\|_g \right] \geq \frac{19}{20} \tag{4}$$

Due to Claim 2.4,

$$\left| \left\{ t_j \;\middle|\; t_j \in T, \|\Delta_{t_j}\|_g \geq \frac{20}{m} \sum_{j'=\frac{m}{100k}}^{m} \|\Delta_{t_{j'}}\|_g \right\} \right| \leq \frac{1}{5} m$$

Combining the above equation with the pigeonhole principle, for any $I \subseteq T$ with $|I| \geq m/4$, we have

$$\left| \left\{ t_j \;\middle|\; t_j \in I, \|\Delta_{t_j}\|_g < \frac{20}{m} \sum_{j'=\frac{m}{100k}}^{m} \|\Delta_{t_{j'}}\|_g \right\} \right| \geq \frac{1}{4} m - \frac{1}{5} m = \frac{1}{20} m \tag{5}$$

Consider the quantity $\|\Delta_j\|_g + \sum_{j'=1}^{|H|} \|(\Delta_H)_{j'}\|_g$ in Eq. (3). We use Eq. (4) and Eq. (5) to provide an upper bound,

$$\|\Delta_j\|_g + \sum_{j'=1}^{|H|} \|(\Delta_H)_{j'}\|_g \leq \left( \frac{20}{m} + \frac{400k}{m} \right) \sum_{j'=\frac{m}{100k}}^{m} \|\Delta_{t_{j'}}\|_g.$$

Eq. (4) will decrease the final probability by $(1 - 19/20)$ (from $1/4$ to $1/4 - 1/20$). Eq. (5) will decrease the size of this set of $j$ by $\frac{1}{5} m$ (from $\frac{1}{4} m$ to $\frac{1}{4} m - \frac{1}{5} m$).

Putting it all together, we can update Eq. (3) in the following sense,

$$\Pr_{H \sim \binom{T}{2k}} \left[ \left| \left\{ j \;\middle|\; j \in T, \; \min_{x \in \mathbb{R}^{|H|}} \|A_H x - A_j\|_g \leq C \cdot \frac{1}{m} \cdot \sum_{j'=\frac{m}{100k}}^{m} \|\Delta_{t_{j'}}\|_g \right\} \right| \geq (\frac{1}{4} - \frac{1}{5}) m \right] \geq \frac{1}{4} - \frac{1}{20}.$$

where

$$C = (400 + 20) \cdot k \cdot \mathrm{ati}_{g,|H|+1} \cdot \mathrm{mon}_g.$$

$\square$

## A.6  Proof of Theorem 1.2

*Proof.* The running time is discussed at the beginning of Section 2. In the remaining of the proof, we will focus on the correctness of Algorithm 1.

Firstly, let us consider the size of the output $S$. For $i \in \{0\} \cup [r]$, let $m_i = |T_i|$. We set number of rounds $r$ to be the smallest value such that $m_r < 1000k$. By the algorithm, we have $m_i = m_{i-1} - 2k - (m_{i-1} - 2k)/20 \leq 19/20 \cdot m_{i-1}$. Thus, $r = O(\log n)$. In each round $i$, the size of $S_i$ is $2k$. Then $|S| = |T_r| + \sum_{i=1}^{r} |S_i| \leq 1000k + r \cdot 2k \leq O(k \log n)$.

Next, let us consider the quality of $S$. Since each regression call has $1 - 1/\mathrm{poly}(n)$ success probability, all the regression calls succeed with probability at least $1 - 1/\mathrm{poly}(n)$. In the remaining of the proof, we condition on that all the regression calls succeed.

Let us fix $i \in [r], j \in [\log n]$. Recall that $T_i = \{t_{i,1}, t_{i,2}, \cdots, t_{i,m_i}\}$ and $\|\Delta_{t_{i,1}}\|_g \geq \|\Delta_{t_{i,2}}\|_g \geq \cdots \geq \|\Delta_{t_{i,m_i}}\|_g$. By regression property and Lemma 2.5, with probability at least $1/4$,

$$\sum_{q \in R_i^{(j)}} \min_{x \in \mathbb{R}^{2k}} \|A_{S_i^{(j)}} x - A_q\|_g$$

$$\leq \sum_{q \in R_i^{(j)}} v_{i,q}^{(j)}$$

$$\leq \text{reg}_{g,2k} \cdot (m_i - 2k) \cdot 500 \cdot k \cdot \text{ati}_{g,2k+1} \cdot \text{mon}_g / m_i \cdot \sum_{j' = \frac{m_i}{100k}}^{m_i} \|\Delta_{t_{i,j'}}\|_g$$

$$\leq \text{reg}_{g,2k+1} \cdot \text{ati}_{g,2k+1} \cdot \text{mon}_g \cdot O(k) \cdot \sum_{j' = \frac{m_i}{100k}}^{m_i} \|\Delta_{t_{i,j'}}\|_g.$$

For each $i \in [r]$, since we repeat $\log(n)$ times, the success probability can be boosted to at least $1 - 1/\text{poly}(r)$, i.e., with probability at least $1 - 1/\text{poly}(r)$, we have

$$\sum_{q \in R_i} \min_{x \in \mathbb{R}^{2k}} \|A_{S_i} x - A_q\|_g \leq \text{reg}_{g,2k+1} \cdot \text{ati}_{g,2k+1} \cdot \text{mon}_g \cdot O(k) \cdot \sum_{j' = \frac{m_i}{100k}}^{m_i} \|\Delta_{t_{i,j'}}\|_g. \quad (6)$$

In the remaining of the proof, we condition on above inequality for every $i \in [r]$. Without loss of generality, we suppose $\|\Delta_1\|_g \geq \|\Delta_2\|_g \geq \cdots \geq \|\Delta_n\|_g$. We have

$$\sum_{q=1}^{n} \min_{x \in \mathbb{R}^{|S|}} \|A_S x - A_q\|_g$$

$$\leq \left( \sum_{q \in T_r} \min_{x \in \mathbb{R}^{|T_r|}} \|A_{T_r} x - A_q\|_g \right) + \left( \sum_{i=1}^{r} \sum_{q \in T_{i-1} \setminus T_i} \min_{x \in \mathbb{R}^{2k}} \|A_{S_i} x - A_q\|_g \right)$$

$$= \sum_{i=1}^{r} \sum_{q \in T_{i-1} \setminus T_i} \min_{x \in \mathbb{R}^{2k}} \|A_{S_i} x - A_q\|_g$$

$$\leq \sum_{i=1}^{r} \text{reg}_{g,2k+1} \cdot \text{ati}_{g,2k+1} \cdot \text{mon}_g \cdot O(k) \cdot \sum_{j' = \frac{m_i}{100k}}^{m_i} \|\Delta_{t_{i,j'}}\|_g$$

$$\leq \sum_{i=1}^{r} \text{reg}_{g,2k+1} \cdot \text{ati}_{g,2k+1} \cdot \text{mon}_g \cdot O(k) \cdot \sum_{j' = \frac{m_i}{100k}}^{m_i} \|\Delta_{j'}\|_g$$

$$= \text{reg}_{g,2k+1} \cdot \text{ati}_{g,2k+1} \cdot \text{mon}_g \cdot O(k) \cdot \sum_{j'=1}^{n} \|\Delta_{j'}\|_g \left( \operatorname*{argmin}_{i \in [r]} \{m_i < j'\} - \operatorname*{argmin}_{i \in [r]} \left\{ \frac{m_i}{100k} < j' \right\} + O(1) \right)$$

$$\leq \text{reg}_{g,2k+1} \cdot \text{ati}_{g,2k+1} \cdot \text{mon}_g \cdot O(k \log k) \cdot \|\Delta\|_g,$$

where the third step follows from $T_{i-1} \setminus T_i = S_i \cup R_i$ and Equation (6), the forth step follows from $\|\Delta_{j'}\|_g \geq \|\Delta_{t_{i,j'}}\|_g$, and the last step follows from $\left( \operatorname*{argmin}_{i \in [r]} \{m_i < j'\} - \operatorname*{argmin}_{i \in [r]} \left\{ \frac{m_i}{100k} < j' \right\} + O(1) \right) \leq O(\log k)$.

$\square$

# B   Necessity of the Properties of $g$

We note that an approximate triangle inequality is necessary to obtain a column subset selection algorithm. An example function not satisfying this is the "jumping function": $g_\tau(x) = |x|$ if $|x| \geq \tau$, and $g_\tau(x) = 0$ otherwise. For the identity matrix $I$ and any $k = \Omega(\log n)$, the Johnson-Lindenstrauss lemma implies one can find a rank-$k$ matrix $B$ for which $\|I - B\|_\infty < 1/2$, that is, all entries of

$I - B$ are at most $1/2$. If we set $\tau = 1/2$, then $\|I - B\|_{g_\tau} = 0$, but for any subset $I_S$ of columns of the identity matrix we choose, necessarily $\|I - I_S X\|_\infty \geq 1$, so $\|I - B\|_{g_\tau} > 0$. Consequently, there is no subset of a small number of columns which obtains a $\mathrm{poly}(k \log n)$-approximation with the jumping function loss measure.

While the jumping function does not satisfy the Approximate triangle inequality, it does satisfy our only other required structural property, the Monotone property.

There are interesting examples of functions $g$ which are only approximately monotone in the above sense, such as the quantile function $\rho_\tau(x)$, studied in [39] in the context of regression, where for a given parameter $\tau$, $\rho_\tau(x) = \tau x$ if $x \geq 0$, and $\rho_\tau(x) = (\tau - 1)x$ if $x < 0$. Only when $\tau = 1/2$ is this a monotone function with $\mathrm{mon}_g = 1$ in the above definition, in which case it coincides with the absolute value function up to a factor of $1/2$. For other constant $\tau \in (0, 1)$, $\mathrm{mon}_g$ is a constant. The loss function $\rho_\tau(x)$ is also sometimes called the scalene loss, and studied in the context of low rank approximation in [13].

When $\tau = 1$ this is the so-called Rectified Linear Unit (ReLU) function in machine learning, i.e., $\rho_1(x) = x$ if $x \geq 0$ and $\rho_1(x) = 0$ if $x < 0$. In this case $\mathrm{mon}_g = \infty$. and the optimal rank-$k$ approximation for any matrix $A$ is 0, since $\|A - \lambda \mathbf{1}\mathbf{1}^\top\|_{\rho_1} = 0$ if one sets $\lambda$ to be a large enough positive number, thereby making all entries of $A - \lambda \mathbf{1}\mathbf{1}^\top$ negative and their corresponding cost equal to 0. Notice though, that there are no good column subset selection algorithms for some matrices $A$, such as the $n \times n$ identity matrix. Indeed, for the identity, if we choose any subset $A_S$ of at most $n - 1$ columns of $A$, then for any matrix $X$ there will be an entry of $A - A_S X$ which is positive, causing the cost to be positive. Since we will restrict ourselves to column subset selection, being approximately monotone with a small value of $\mathrm{mon}_g$ in the above definition is in fact necessary to obtain a good approximation with a small number of columns, as the ReLU function illustrates (see also related functions such as the leaky ReLU and squared ReLU [40, 41, 42]).

Note that the ReLU function is an example which satisfies the triangle inequality, showing that our additional assumption of approximate monotonicity is required.

Thus, if either property fails to hold, there need not be a small subset of columns spanning a relative error approximation. These examples are stated in more detail below.

## B.1 Functions without Approximate Triangle Inequality

In this section, we show how to construct a function $f$ such that it is not possible to obtain a good entrywise-$f$ low rank approximation by selecting a small subset of columns. Furthermore, $f$ is monotone but does not have the approximate triangle inequality. Theorem B.4 shows this result.

First, we show that a small subset of columns cannot give a good low rank approximation in $\ell_\infty$ norm. Then we reduce the $\ell_\infty$ column subset selection problem to the entrywise-$f$ column subset selection problem.

The following is the Johnson-Lindenstrauss lemma.

**Lemma B.1** (JL Lemma). *For any $n \geq 1, \epsilon \in (1/\sqrt{n}, 1/2)$, there exists $U \in \mathbb{R}^{n \times k}$ with $k = O(\epsilon^{-2} \log(n))$ such that $\|UU^\top - I_n\|_\infty \leq O(\epsilon)$, where $I_n \in \mathbb{R}^{n \times n}$ is an identity matrix.*

**Theorem B.2.** *For $n \geq 1$, there is a matrix $A \in \mathbb{R}^{n \times n}$ with the following properties. Let $k = \Theta(\epsilon^{-2} \log(n))$ for an arbitrary $\epsilon \in (1/\sqrt{n}, 1/2)$. Let $D \in \mathbb{R}^{n \times n}$ denote a diagonal matrix with $n - 1$ nonzeros on the diagonal. We have*

$$\min_{X \in \mathbb{R}^{n \times n}} \|XDA - A\|_\infty \geq 1$$

*and*

$$\min_{\mathrm{rank} -k \ A'} \|A' - A\|_\infty < O(\epsilon).$$

*Proof.* We choose $A$ to be the identity matrix. By Lemma B.1, we can find a rank-$k$ matrix $B$ for which

$$\|A - B\|_\infty \leq O(\epsilon).$$

Since $A$ is an $n \times n$ identity matrix, even if we can use $n - 1$ columns to fit the other columns, the cost is still at least 1. $\square$

In the following, we state the construction of our function $f$.

**Definition B.3.** *We define function $f(x)$ to be $f(x) = c$ if $|x| > \tau$ and $f(x) = 0$ if $|x| \leq \tau$. Given matrix A, we define $\|A\|_f = \sum_{i=1}^{n} \sum_{j=1}^{n} f(A_{i,j})$.*

**Theorem B.4** (No good subset of columns). *For any $n \geq 1$, there is a matrix $A \in \mathbb{R}^{n \times n}$ with the following property. Let $k \geq c \log n$ for a sufficiently large constant $c > 0$. Let $D \in \mathbb{R}^{n \times n}$ denote an arbitrary diagonal matrix with $n - 1$ nonzeros on the diagonal. For $f$ with parameter $\tau = 1/4$, we have*

$$\min_{X \in \mathbb{R}^{n \times n}} \|XDA - A\|_f > 0$$

*and*

$$\min_{\text{rank} - k \ A'} \|A' - A\|_f = 0.$$

*Proof.* We can set $A$ to be the identity matrix. Due to Theorem B.2, there exists $A'$ for which $\min_{\text{rank} - k \ A'} \|A' - A\|_\infty < 1/4$, which implies that $\min_{\text{rank} - k \ A'} \|A' - A\|_f = 0$. Also due to Theorem B.2, we have $\min_{X \in \mathbb{R}^{n \times n}} \|XDA - A\|_\infty = 1$, and thus, $\min_{X \in \mathbb{R}^{n \times n}} \|XDA - A\|_f > 0$. $\square$

## B.2 ReLU **Function Low Rank Approximation**

In this section, we discuss a function which has the approximate triangle inequality but is not monotone. The specific function we discuss in this section is ReLU . The definition of ReLU is defined in Definition B.5. First, we show that ReLU low rank approximation has a trivial best rank-$k$ approximation. Second, we show that for some matrices, there is no small subset of columns which can give a good low rank approximation.

**Definition B.5.** *We define function $\text{ReLU}(x)$ to be $\text{ReLU}(x) = \max(0, x)$. Given matrix A, we define $\|A\|_{\text{ReLU}} = \sum_{i=1}^{n} \sum_{j=1}^{n} \text{ReLU}(A_{i,j})$.*

In the rank-$k$ approximation problem, given an input matrix $A$, the goal is to find a rank-$k$ matrix $B$ for which $\|A - B\|_{\text{ReLU}}$ is minimized. A simple observation is that if we set $B$ to be a matrix with each entry of value $\|A\|_\infty$, then the value of each entry of $A - B$ is at most 0. Thus, $\|A - B\|_{\text{ReLU}} = 0$. Furthermore, the rank of $B$ is 1.

Now, consider the column subset selection problem, let input matrix $A \in \mathbb{R}^{n \times n}$ be an identity matrix. Then even if we can choose $n - 1$ columns, they can never fit the remaining column. Thus, the cost is at least 1. But as discussed, the best rank-$k$ cost is always 0. This implies that any subset of columns cannot give a good rank-$k$ approximation.

## C   Regression Solvers

In this section, we discuss several regression solvers.

### C.1   Regression for Convex $g$

Notice that when the function $g$ is convex, the regression problem $\min_{X \in \mathbb{R}^{d \times m}} \|AX - B\|_g$ for any given matrices $A \in \mathbb{R}^{n \times d}, B \in \mathbb{R}^{n \times m}$ is a convex optimization problem. Thus, it can be solved exactly by convex optimization algorithms.

**Fact C.1.** *Let $g$ be a convex function. Given $A \in \mathbb{R}^{n \times d}, B \in \mathbb{R}^{n \times m}$, the regression problem $\min_{X \in \mathbb{R}^{d \times m}} \|AX - B\|_g$ can be solved exactly by convex optimization in $\text{poly}(n, d, m)$ time.*

If a function $g$ has additional properties, i.e. $g$ is symmetric, monotone and grows subquadratically, then there is a better running time constant approximation algorithm shown in [43]. Here "grows quadratically" means that there is an $\alpha \in [1, 2]$ and $c_g > 0$ so that for $a, a'$ with $|a| > |a'| > 0$,

$$\left|\frac{a}{a'}\right|^\alpha \geq \frac{g(a)}{g(a')} \geq c_g \left|\frac{a}{a'}\right|.$$

This kind of function $g$ is also called a "sketchable" function. Notice that the Huber function satisfies the above properties.

**Theorem C.2** (Modified version of Theorem 3.1 of [43]). *Function $g$ is symmetric, monotone and grows subquadratically ($g$ is a G-function defined by [43]). Given a matrix $A \in \mathbb{R}^{n \times d}$ and a matrix $B \in \mathbb{R}^{n \times m}$, there is an algorithm which can output a matrix $\widehat{X} \in \mathbb{R}^{d \times m}$ and a fitting cost vector $y \in \mathbb{R}^m$ such that with probability at least $1 - 1/\operatorname{poly}(nm)$, $\forall i \in [m], \|A\widehat{X}_i - B_i\|_g \leq O(1) \cdot \min_{x \in \mathbb{R}^d} \|Ax - B_i\|_g$, and $y_i = \Theta(\|A\widehat{X}_i - B_i\|_g)$. Furthermore, the running time is at most $\widetilde{O}(\operatorname{nnz}(A) + \operatorname{nnz}(B) + m \cdot \operatorname{poly}(d \log n))$.*

*Proof.* We run $O(\log(nm))$ repetitions of the single column regression algorithm shown in Theorem 3.1 of [43] for all columns $B_i$ for $i \in [m]$. For each regression problem $\|Ax - B_i\|_g$, we take the solution whose estimated cost is the median among these $O(\log(nm))$ repetitions as $\widehat{X}_i$. Then by the Chernoff bound, we can boost the success probability of each column to $1 - 1/\operatorname{poly}(nm)$. By taking a union bound over all columns, we complete the proof. $\square$

## C.2 $\ell_p$ Regression

One of the most important cases in regression and low rank approximation problems is when the error measure is $\ell_p$. For $\ell_p$ regression, though it can be solved by convex optimization/linear programming exactly, we can get a much faster running time if we allow some approximation ratios. In the following theorem, we show that there is an algorithm which can be used to solve $\ell_p$ regression for any $p \geq 1$.

**Theorem C.3** (Modified version of [44]). *Let $p \geq 1, \epsilon \in (0, 1)$. Given a matrix $A \in \mathbb{R}^{n \times d}$ and a matrix $B \in \mathbb{R}^{n \times m}$, there is an algorithm which can output a matrix $\widehat{X} \in \mathbb{R}^{d \times m}$ and a fitting cost vector $y \in \mathbb{R}^m$ such that with probability at least $1 - 1/\operatorname{poly}(nm)$, $\forall i \in [m], \|A\widehat{X}_i - B_i\|_p^p \leq (1 + \epsilon) \cdot \min_{x \in \mathbb{R}^d} \|Ax - B_i\|_p^p$, and $y_i = \Theta(\|A\widehat{X}_i - B_i\|_p^p)$. Furthermore, the running time is at most $\widetilde{O}(\operatorname{nnz}(A) + \operatorname{nnz}(B) + mn^{\max(1-2/p,0)} \cdot \operatorname{poly}(d))$.*

*Proof.* As in the proof of Theorem C.2, we only need to run $O(\log(nm))$ repetitions of the single column regression algorithm shown in [44]. $\square$

## C.3 $\ell_0$ Regression

**Definition C.4** (Regular partition). *Given a matrix $A \in \mathbb{R}^{n \times k}$, we say $\{S_1, S_2, \cdots, S_h\}$ is a regular partition for $[n]$ with respect to the matrix $A$ if, for each $i \in [h]$,*

$$\operatorname{rank}(A^{S_i}) = |S_i|, \text{ and } \operatorname{rowspan}(A^{S_i}) = \operatorname{rowspan}\left(A^{\cup_{j=i}^h S_j}\right),$$

*where $A^{S_i} \in \mathbb{R}^{|S_i| \times k}$ denotes the matrix that selects a subset $S_i$ of rows of the matrix $A$.*

---

**Algorithm 2** $\ell_0$ regression [28]

**procedure** L0REGRESSION$(A, b, n, k, c)$          $\triangleright$ Theorem C.5
    $x' \leftarrow 0^k$
    $\{S_1, S_2, \cdots, S_h\} \leftarrow$ GENERATEREGULARPARTITION$(A, n, k)$
    $x' \leftarrow 0^k$
    **for** $i = 1 \to h$ **do**
        Find a $\widetilde{x}$ such that $A^{S_i}\widetilde{x} = b_{S_i}$
        **if** $\|A\widetilde{x} - b\|_0 < \|Ax' - b\|_0$ **then**
            $x' \leftarrow \widetilde{x}$
        **end if**
    **end for**
    **return** $x'$
**end procedure**

**Theorem C.5** (Generalization of [28]). *Given matrix $A \in \mathbb{R}^{n \times k}$ and vector $\mathbb{R}^n$, for any $c \in [1, k]$, there is an algorithm (Algorithm 2) that runs in $n^{O(1)}$ time and outputs a vector $x' \in \mathbb{R}^k$ such that*

$$\|Ax' - b\|_0 \le k \min_{x \in \mathbb{R}^k} \|Ax - b\|_0.$$

*Proof.* Let $x^* \in \mathbb{R}^k$ denote the optimal solution to $\min_{x \in \mathbb{R}^k} \|Ax - b\|_0$. We define set $E$ as follows

$$E = \{i \in [n] \mid (Ax^*)_i \ne b_i\}.$$

We create a regular partition $\{S_1, S_2, \cdots, S_h\}$ for $[n]$ with respect to $A$.

Let $i$ denote the smallest index such that $|S_i \cap E| = 0$, i.e.,

$$i = \min\{j \mid |S_j \cap E| = 0\}.$$

The linear equation we want to solve is $A^{S_i} x = b_{S_i}$. Let $\widetilde{x} \in \mathbb{R}^k$ denote a solution to $A^{S_i} \widetilde{x} = A^{S_i} x^*$ (Note that, by our choice of $i$, $b_{S_i} = A^{S_i} x^*$). Then we can rewrite $\|A\widetilde{x} - b\|_0$ in the following sense,

$$\|A\widetilde{x} - b\|_0 = \sum_{j=1}^{i-1} \left\| A^{S_j} \widetilde{x} - b_{S_j} \right\|_0 + \sum_{j=i}^{h} \left\| A^{S_j} \widetilde{x} - b_{S_j} \right\|_0. \tag{7}$$

For each $j \in \{1, 2, \cdots, i-1\}$, we have

$$\begin{aligned} \|A^{S_j} \widetilde{x} - b_{S_j}\|_0 &\le k \\ &\le \|A^{S_j} x^* - b_{S_j}\|_0 \cdot \lceil k \rceil, \end{aligned} \tag{8}$$

where the first step follows from $|S_0| \le k$, and the last step follows from $\|A^{S_j} x^* - b_{S_j}\|_0 \ge 1$, $\forall j \in [i-1]$.

Note that, by our choice of $i$, we have $A^{S_i} \widetilde{x} = A^{S_i} x^*$. Then for each $j \in \{i, i+1, \cdots, n\}$, using the regular partition property, there always exists a matrix $P_{(j)}$ such that $A^{S_j} = P_{(j)} A^{S_i}$. Then we have

$$A^{S_j} \widetilde{x} = P_{(j)} A^{S_i} \widetilde{x} = P_{(j)} A^{S_i} x^* = A^{S_j} x^*. \tag{9}$$

Plugging Eq. (8) and (9) into Eq. (7), we have

$$\begin{aligned} \|A\widetilde{x} - b\|_0 &= \sum_{j=1}^{i-1} \left\| A^{S_j} \widetilde{x} - b_{S_j} \right\|_0 + \sum_{j=i}^{h} \left\| A^{S_j} \widetilde{x} - b_{S_j} \right\|_0 \\ &\le k \sum_{j=1}^{i-1} \left\| A^{S_j} x^* - b_{S_j} \right\|_0 + \sum_{j=i}^{h} \left\| A^{S_j} x^* - b_{S_j} \right\|_0 \\ &\le k \|Ax^* - b\|_0. \end{aligned}$$

This completes the proof. □

# D Hardness

## D.1 Column Subset Selection for the Huber Function

The rough idea here is to define $k = \Omega(\sqrt{\log n})$ groups of columns, where we carefully choose the $i$-th group to have $n^{1-2i\epsilon}$ columns, $\epsilon = .2/(1.5k)$, and in the $i$-th group each column has the form

$$n^{1.5i\epsilon} \cdot 1^n + [\pm n^{-.2+i\epsilon}, \ldots, \pm n^{-.2+i\epsilon}, \pm n^{.5+2i\epsilon}, \ldots, \pm n^{.5+2i\epsilon}],$$

where there are $n - n^{.1}$ coordinates where the perturbation is randomly either $+n^{-.2+i\epsilon}$ or $-n^{-.2+i\epsilon}$, and the remaining $n^{.1}$ coordinates are randomly either $+n^{.5+2i\epsilon}$ or $-n^{.5+2i\epsilon}$. We call the former type of coordinates "small noise", and the latter "large noise". All remaining columns in the matrix are set to $0$. Because of the random signs, it is very hard to fit the noise in one column to that of another column. One can show, that to approximate a column in the $j$-th group by a column in the $i$-th group, $i < j$, one needs to scale by roughly $n^{1.5(j-i)\epsilon}$, just to cancel out the "mean" $n^{1.5j\epsilon} \cdot 1^n$.

But when doing so, since the Huber function is quadratic for small values, the scaled small noise is now magnified more than linearly compared to what it was before, and this causes a column in the $i$-th group not to be a good approximation of a column in the $j$-th group. On the other hand, if you want to approximate a column in the $j$-th group by a column in the $i$-th group, $i > j$, one again needs to scale by roughly $n^{1.5(j-i)\epsilon}$ just to cancel out the "mean", but now one can show the large noise from the column in the $i$-th group is too large and remains in the linear regime, causing a poor approximation. The details of this construction are given in the following theorem.

**Theorem D.1.** *Let $H(x)$ denote the modified Huber function with $\tau = 1$, i.e.,*

$$H(x) = \begin{cases} x^2/\tau, & \text{if } |x| < \tau; \\ |x|, & \text{if } |x| \geq \tau. \end{cases}$$

*For any $n \geq 1$, there is a matrix $A \in \mathbb{R}^{n \times n}$ such that, if we select $o(\sqrt{\log n})$ columns to fit the entire matrix, there is no $O(1)$-approximation, i.e., for any subset $S \subseteq [n]$ with $|S| = o(\sqrt{\log n})$,*

$$\min_{X \in \mathbb{R}^{|S| \times n}} \|A_S X - A\|_H \geq \omega(1) \cdot \min_{\text{rank} -1 \; A'} \|A' - A\|_H.$$

*Proof.* Suppose there is an algorithm which only finds a subset with size $k/2 = o(\sqrt{\log n})$. We want to prove a lower bound on its approximation ratio.

Let $\epsilon = 0.2/(1.5k)$. Let $A$ denote a matrix with $k + 1$ groups of columns.

For each group $i \in [k]$, $I_i$ has $n^{1-2i\epsilon}$ columns which are

$$\begin{bmatrix} n^{1.5i\epsilon} \\ n^{1.5i\epsilon} \\ n^{1.5i\epsilon} \\ \vdots \\ n^{1.5i\epsilon} \\ n^{1.5i\epsilon} \\ n^{1.5i\epsilon} \\ n^{1.5i\epsilon} \end{bmatrix} + \begin{bmatrix} \pm n^{-0.2+i\epsilon} \\ \pm n^{-0.2+i\epsilon} \\ \vdots \\ \pm n^{-0.2+i\epsilon} \\ \pm n^{0.5+2i\epsilon} \\ \pm n^{0.5+2i\epsilon} \\ \vdots \\ \pm n^{0.5+2i\epsilon} \end{bmatrix} \in \mathbb{R}^n,$$

where $\pm$ indicates i.i.d. random signs. For the error column, the first $n - n^{0.1}$ rows are $n^{-0.2+i\epsilon}$, and the last $n^{0.1}$ rows are $n^{0.5+2i\epsilon}$.

The last group of $n - \sum_{i=1}^{k} n^{1-2i\epsilon}$ columns are

$$\begin{bmatrix} 0 \\ 0 \\ \vdots \\ 0 \\ 0 \end{bmatrix} \in \mathbb{R}^n.$$

The optimal cost is at most

$$\sum_{i=1}^{k} n^{1-2i\epsilon} \cdot \left( (n - n^{0.1}) H(n^{-0.2+i\epsilon}) + n^{0.1} H(n^{0.5+2i\epsilon}) \right) \leq \sum_{i=1}^{k} n^{1-2i\epsilon} (n \cdot n^{-0.4+2i\epsilon} + n^{0.6+2i\epsilon}) \leq O(kn^{1.6}).$$

where the second step follows since $n^{-0.2+i\epsilon} < 1$ and $n^{0.5+2i\epsilon} \geq 1$. Thus, it implies

$$\min_{\text{rank} -1 \; A'} \|A' - A\|_H \leq O(kn^{1.6}).$$

Now let us consider the lower bound for using a subset of columns to fit the matrix. First, we fix a set $S = \{j_1, j_2, \cdots, j_{k/2}\}$ of $k/2$ columns. Since there are $k$ groups, and $|S| \leq k/2$, the number of groups $I_i$ for $i \in [k]$ with $S \cap I_i = \emptyset$ is at least $k/2$. It means that there are at least $k/2$ groups for which $S$ does not have any column from them. Notice that the optimal cost is at most $O(kn^{1.6})$, so it suffices to prove that $\forall i \in [k]$ with $I_i \cap S = \emptyset$, each column $j \in I_i$ will contribute a cost of $\omega(n^{0.6+2i\epsilon})$.

For notation, we use $\text{group}(j)$ to denote the index of the group which contains the column $j$. For each column $j$, we use $\Delta_j$ to denote the noise part, and use $A_j^*$ to denote the rank-1 "ground truth" part. Notice that $A_j = A_j^* + \Delta_j$.

**Claim D.2** (Noise cannot be used to fit other vectors). *Let $x_1, x_2, \cdots, x_s \in \mathbb{R}^m$ be $s$ random sign vectors. Then with probability at least $1 - 2^s \cdot 2^{-\Theta(m/2^s)}$, $\forall \alpha_1, \alpha_2, \cdots, \alpha_s \in \mathbb{R}$, the size of $Z_{\alpha_1, \alpha_2, \cdots, \alpha_s} = \{i \in [m] \mid \text{sign}((\alpha_1 x_1)_i) = \text{sign}((\alpha_2 x_2)_i) = \cdots = \text{sign}((\alpha_s x_s)_i) = \text{sign}(+1)\}$ is at least $\Omega(m/2^s)$.*

*Proof.* For a set of fixed $\alpha_1, \alpha_2, \cdots, \alpha_s$, the claim follows from the Chernoff bound. Since there are $2^s$ different possibilities of signs of $\alpha_1, \cdots, \alpha_s$, taking a union bound over them completes the proof. $\qquad\square$

Now we consider a specific column $j \in I_i$ for some $i \in [k]$, where $I_i \cap S = \emptyset$. Suppose the fitting coefficients are $\alpha_1, \alpha_2, \cdots, \alpha_{k/2}$. Consider the following term

$$(\alpha_1 A_{j_1} + \alpha_2 A_{j_2} + \cdots + \alpha_{k/2} A_{k/2}) - A_j$$
$$= (\alpha_1 A_{j_1}^* + \alpha_2 A_{j_2}^* + \cdots + \alpha_{k/2} A_{k/2}^* - A_j^*) + (\alpha_1 \Delta_{j_1} + \alpha_2 \Delta_{j_2} + \cdots + \alpha_{k/2} \Delta_{k/2} - \Delta_j)$$

Let $u^* = (\alpha_1 A_{j_1}^* + \alpha_2 A_{j_2}^* + \cdots + \alpha_{k/2} A_{k/2}^* - A_j^*)$. By Claim D.2, with probability at least $1 - (k/2) \cdot 2^{-\Theta(n/2^{k/2})}$,

$$|\{t \in [n] \mid \text{sign}(u_t^*) = \text{sign}(\alpha_1 \Delta_{j_1,t}) = \cdots = \text{sign}(\alpha_{k/2} \Delta_{j_{k/2},t}) = \text{sign}(-\Delta_{j,t})\}| = \Omega(n/2^{k/2}).$$

Observe that all the coordinates of $u^*$ are the same, and the absolute value of each entry of $u^*$ should be at most $O(n^{1.5i\epsilon})$. Otherwise the column already has $\omega(n/2^{k/2} \cdot n^{1.5i\epsilon}) = \omega(n^{0.6+2i\epsilon})$ cost. Thus, the magnitude of each entry of $\alpha_1 A_{j_1}^* + \alpha_2 A_{j_2}^* + \cdots + \alpha_{k/2} A_{k/2}^*$ is $\Theta(n^{1.5i\epsilon})$. Thus, there exists $t \in [k/2]$, such that the absolute value of each entry of $\alpha_t A_{j_t}^*$ is at least $\Omega(n^{1.5i\epsilon}/k)$. Then there are two cases.

The first case is $\text{group}(t) < \text{group}(j)$. Let $\text{group}(t) = i'$. Then $|\alpha_t| = \Omega(n^{1.5(i-i')\epsilon}/k)$. By Claim D.2 again, with probability at least $1 - (k/2) \cdot 2^{-\Theta(n/2^{k/2})}$, the size of

$$\{z \in [n - n^{0.1}] \mid \text{sign}(u_z^*) = \text{sign}(\alpha_1 \Delta_{j_1,z}) = \cdots = \text{sign}(\alpha_{k/2} \Delta_{j_{k/2},z}) = \text{sign}(-\Delta_{j,z})\}$$

is at least $\Omega(n/2^{k/2})$. Thus, the cost to fit is at least $\Omega(n/2^{k/2}) \cdot (n^{-0.2+i'\epsilon} n^{1.5(i-i')\epsilon}/k)^2 = \omega(n^{0.6+2i\epsilon})$.

The second case is $\text{group}(t) > \text{group}(j)$. Let $\text{group}(t) = i'$. Then $|\alpha_t| = \Omega(n^{1.5(i-i')\epsilon}/k)$. By Claim D.2 again, with probability at least $1 - (k/2) \cdot 2^{-\Theta(n^{0.1}/2^{k/2})}$, the size of

$$\{z \in \{n - n^{0.1} + 1, \cdots, n\} \mid \text{sign}(u_z^*) = \text{sign}(\alpha_1 \Delta_{j_1,z}) = \cdots = \text{sign}(\alpha_{k/2} \Delta_{j_{k/2},z}) = \text{sign}(-\Delta_{j,z})\}$$

is at least $\Omega(n^{0.1}/2^{k/2})$. Thus, the fitting cost is at least $\Omega(n^{0.1}/2^{k/2}) \cdot (n^{0.5+2i'\epsilon} n^{1.5(i-i')\epsilon}/k) = \omega(n^{0.6+2i\epsilon})$.

By taking a union bound over all columns $j$, we have with probability at least $1 - 2^{n^{\Theta(1)}}$, the total cost to fit by a column subset $S$ is at least $\omega(kn^{1.6})$.

Then, by taking a union bound over all the $\binom{n}{k}$ number of sets $S$, we complete the proof. $\qquad\square$

## D.2 Column Subset Selection for the Reverse Huber Function

In this section, we consider a "reverse Huber function": $g(x) = x^2$ if $x \geq 1$ and $g(x) = |x|$ for $x \leq 1$.

**Theorem D.3.** *Let $g(x)$ denote the "reverse Huber function" with $\tau = 1$, i.e.,*

$$H(x) = \begin{cases} x^2/\tau, & \text{if } |x| > \tau; \\ |x|, & \text{if } |x| \leq \tau. \end{cases}$$

*For any $n \geq 1$, there is a matrix $A \in \mathbb{R}^{n \times n}$ such that, if we select only $1$ column to fit the entire matrix, there is no $n^{o(1)}$-approximation to the best rank-$1$ approximation, i.e., for any subset $S \subseteq [n]$ with $|S| = 1$,*

$$\min_{X \in \mathbb{R}^{|S| \times n}} \|A_S X - A\|_g \geq n^{\Omega(1)} \cdot \min_{\text{rank}-1 \ A'} \|A' - A\|_g.$$

$$H(x) = \begin{cases} x^2/2\tau & \text{if } |x| \le \tau; \\ |x| - \tau/2 & \text{otherwise.} \end{cases}$$

$$\widetilde{H}(x) = \begin{cases} |x|/2 & \text{if } |x| \le \tau; \\ x^2/4\tau + \tau/4 & \text{otherwise.} \end{cases}$$

$\tau = 0.8$     $\tau = 1$     $\tau = 2$

Figure 3: The blue curve is the Huber function which combines an $\ell_2$-like measure for small $x$ with an $\ell_1$-like measure for large $x$. The red curve is the "reverse" Huber function which combines an $\ell_1$-like measure for small $x$ with an $\ell_2$-like measure for large $x$.

*Proof.* Let $A \in \mathbb{R}^{n \times n}$ have one column that is $a = (n^{1/2}, 0, \ldots, 0)^\top$ and $n-1$ columns that are each equal to $b = (0, 1/n, 1/n, \ldots, 1/n)^\top$. If we choose one column which has the form as $a$ to fit the other columns, the cost is at least $(n-1)^2/n = \Theta(n)$. If we choose one column which has the form as $b$ to fit the other columns, the cost is at least $(n^{1/2})^2 = \Theta(n)$.

Now we consider using a vector $c = (1/n^{1/4}, 1/n, 1/n, \ldots, 1/n)^\top$ to fit all the columns. One can use $c$ to approximate $a$ with cost at most $(n-1) \cdot n^{3/4}/n = \Theta(n^{3/4})$ by matching the first coordinate, while one can use $c$ to approximate $b$ with cost at most $1/n^{1/4}$ by matching the last $n-1$ coordinates, and since there are $n-1$ columns equal to $b$, the overall total cost of using $c$ to approximate matrix $A$ is $\Theta(n^{3/4})$.

$\square$

## Footnotes

[28] studied the Nearest Codeword problem over finite fields $\mathbb{F}_2$. Their proof can be extended to the real field and generalized to Theorem C.5. For completeness, we still provide the proof of the following result.