[Reviews · NeurIPS 2019]

Reviewer 1



Originality: To the best of my knowledge the results are new. The methods used on the other hand are simple and to my understanding relatively standard in the field. Yet, the fact that they work for a broad class of loss functions is interesting. Quality: The statements of the results and proofs appear to be correct. The motivation to study more general functions is clear and the experiment helps significantly with it (I would even discuss about the experiment earlier in the paper). Yet, I dislike the use of necessary" conditions for the approx. monotonicity and approx. triangle inequality. Indeed, at the complete absence of any of the two conditions the authors provide counterexamples. To my understanding that is not enough for a necessary characterization of the error losses: the condition is necessary only if any function (and not at least one function) which does not satisfy this property cannot be approximated in the appropriate sense. Also, I am not sure what is the zero-one law the authors are referring at the title. It will be great if the authors offer an explanation for it. Clarity: For the most part, the paper is well-written and the arguments relatively clean. One comment is that I consider important is to explain for which choice of r Algorithm 1 works for the general theorem (the initialization step says that r is of order log n which is rather confusing: can I choose r to be 0 for example?) Also I consider important the authors to explain the use of Cramer's rule in the beginning of Subsection 2.1. Significance: The paper points to an interesting direction of classifying the loss functions for which an approximation algorithm with appropriate guarantees can be constructed. They provide certain natural sufficient assumptions under which this is happening, which generalize much beyond the standard \ell_p norms. Their direction can definitely impact future theoretical and potentially empirical research.

Reviewer 2



As a caveat, I am not an expert in the literature surrounding low-rank reconstruction, and may not be entirely correct in my evaluation of the originality and significance of the contributions. Originality:This paper builds upon previous work, in particular [62], which developed column-subset selection for low-rank approximation under the l_p norm. This paper expands upon [62], obtaining results for a broader class of functions and furthermore tightening and fixing some results from [62]. These expansions seem very valuable to the machine learning community. However, the authors may want to further motivate their work by providing specific examples of loss functions to which they extend previous theory, and which have found successful applications in machine learning. Quality: This works appears to be of high quality. The authors provide the intuition behind their theoretical contribution as well as the motivation in a way that is very accessible; the results seem theoretically important, and I can imagine several implications for the machine learning field, for example in using low-rank approximations for problems that are prone to outlier data. Clarity: This paper is somewhat clear, but could in my opinion be improved upon in a few ways that I describe in the relevant section. Overall, the authors guide the reader in understanding the motivation and reason for the hypotheses and class of functions that they investigate. Although I have not carefully read over the proofs, I understand the underlying reasoning of this paper. Significance: This paper appears to be significant due to the importance of low-rank approximation and reconstruction in machine learning, from both a computational efficiency and memory usage perspective (among others). *********** Post-rebuttal comments *********** Based on the author rebuttal and reviewer discussions, I recommend the following for the camera-ready version if the paper is accepted: (i) The issue regarding the choice of r in Algorithm 1 should be clarified. It seems that r simply indicates that the loop in Alg. 1 should be run until |T_r| < 1000k, and that the number of loop iterations required is O(log n); I recommend the authors make this fact obvious within the algorithm itself, for example by changing the first for loop to ''while |T_i| < 1000k''. (ii) Following Reviewer 2's review, I recommend the authors change the title. (iii) Similarly, I agree that the approx. monotone/necessary conditions have been proven to be *sufficient* but not *necessary*; the paper should be updated to clarify this.

Reviewer 3



This paper studies a generalized low-rank approximation problem. In this problem, we are given a matrix $A$ and a cost function $g(.)$. The goal is to find a rank $k$ matrix $B$ such that $\sum_{i, j}g(A_{i,j} - B_{i, j})$ is minimized. In the approximation version of the problem, we aim to find a rank $k polylog (n, k)$ matrix $B'$ so that $\sum_{i, j}g(A_{i,j} - B'_{i, j})$ is a constant times worse than $\sum_{i, j}g(A_{i,j} - B_{i, j})$. The paper's main goal is to understand what kind of $g$ admits approximation solutions. It focuses on column selection algorithms, i.e., the output $B'$ is a linear combination of a subset of columns from $A$. The paper gives a tight characterization of $g(.)$, i.e., it needs to satisfy approximate triangle inequality, and monotone property. Their upper bound result is a generalization of [62]. [62] states that if the cost is $\ell_p$ norm, then there is a good subset of columns to approximate $A$. Their major observations are that (i) [62]'s technique can be used to build a recursive method to find columns that can approximately span $A$, and (ii) [62] is a general technique so many assumptions made there can be relaxed. The authors showed that using only triangle inequality and monotone property suffices to generalize [62]. For the lower bound analysis, the authors claim that both approximate triangle inequality and monotone properties are necessary but the paper has only one lower bound result for the function $H(x)$. They argue that when the cost function is $H(x)$, we can construct a matrix $A$ so that it is impossible to choose a small number of columns to approximate $A$. I think the lower bound result is weaker than what the authors advertised. The generalized low rank approximation problem is an important problem in statistics. The authors gave a non-trivial generalization of [62]. The analysis looks believable. I have two major concerns: 1. Lower bound: I am not able to see why Theorem 1.3 implies that both triangle inequality and monotone property is necessary. 2. Presentation: the authors made a serious effort to explain the intuition for the analysis (Sec 1.1.2) but some of the intuitions still appear to be incomprehensible (e.g., line 158 to line 160; line 168 to line 179). In addition, I feel some of the key definitions are not explained properly. For example, for the equation between line 232 and 233, I cannot understand how Q is defined (e.g., is Q a variable to be optimized or we need a worst-case Q). My concerns could be fixable; I am willing to change my rating if the above two questions are properly addressed/answered.

[Author Response · NeurIPS 2019]

We are thankful to the reviewers for their insightful comments and suggestions. Below we address each reviewer's questions and concerns, with the questions and concerns briefly rephrased in blue.

**Response to Reviewer #2.**

[Choice of $r$]: $r$ is $\Theta(\log n)$. Actually, we do not really need to choose $r$. In Algorithm 1, we drop $1/20$ fraction of $T_{i-1}$ to get $T_i$ in the $i$-th iteration of the outer loop. Thus, the outer loop will have at most $O(\log n)$ iterations. We use $r$ to simplify the notations in our analysis.

[Use of Cramer's rule]: Consider a rank $k$ matrix $M \in \mathbb{R}^{n \times (k+1)}$. Let $P \subseteq [k+1], Q \subseteq [n], |P| = |Q| = k$ be such that $|\det(M_P^Q)|$ is maximized. Since $M$ has rank $k$, we know $\det(M_P^Q) \neq 0$ and thus the columns of $M_P$ are independent. Let $i \in [k+1] \setminus P$. Then the linear equation $M_P x = M_i$ is feasible and there is a unique solution $x$. Furthermore, by Cramer's rule $x_j = \frac{\det(M_{[k+1]\setminus\{j\}}^Q)}{\det(M_P^Q)}$. Since $|\det(M_P^Q)| \geq |\det(M_{[k+1]\setminus\{j\}}^Q)|$, we have $\|x\|_\infty \leq 1$.

We just realized that there is a typo in the equation of line 232, and we correct it as the following:

$$R_{M^*}(H) = \arg \max_{P:P \subseteq H} \left\{ \left| \det\left((M^*)_P^Q\right) \right| \; \middle| \; |P| = |Q| = \text{rank}(M_H^*), Q \subseteq [n] \right\}.$$

["Necessary" conditions and zero-one law]: We can change the wording, and indeed what we meant is that if we're missing any one of the mentioned properties, then we can find an example function for which there is no good column subset selection. On the other hand, if we have all the properties, we have a good column subset selection. This is what we meant in the title by zero-one law, though we are happy to change the wording. We believe this is sometimes loosely what is intended by the word characterization, as also stated by reviewer 4 - that "it needs to satisfy approximate triangle inequality, and monotone property", otherwise there are counterexamples. But, again, we would change the wording.

**Response to Reviewer #3.**

[Experiments on other loss functions]: We are glad to report results with other loss functions in the final version. Due to the page limit of the response, here we only present the result with the loss function "$\ell_1 - \ell_2$" (i.e., $2(\sqrt{1 + x^2/2} - 1)$). The setting is the same as the experiments represented by Figure 2 except that the loss function used is now "$\ell_1 - \ell_2$".

[Minor question, choice of $r$]: See the response to Reviewer #2.

[Textual clarity]: We thank the reviewer for the comments on the presentation.

**Response to Reviewer #4.**

[Lower bound]: We discuss the necessity of the triangle inequality and monotone property in Appendix B. In Appendix B.1, we show that there is no good column subset selection for the jumping function which is monotone but does not satisfy the approximate triangle inequality. In Appendix B.2, we show that there is no good column subset selection for the ReLU function which has the triangle inequality but does not satisfy approximate monotonicity. We can put both results into the main body in the camera ready version.
[Presentation]: We thank the reviewer for the comments on the presentation. We will fix these issues in the camera ready version.

- In the equation between line 232 and line 233, the max is over all possible choices of $P$ and $Q$, and $R_{M^*}(H)$ only takes the value of the corresponding $P$.

- Lines 158 to 160: if we first choose a random subset $T$ of $k$ columns and then randomly choose another column $i$, then $T \cup \{i\}$ is a random set of $k+1$ columns. Consider the submatrix $A_{T \cup \{i\}}$. If we want to choose a subset of $k$ columns from $A_{T \cup \{i\}}$ to approximate $A_{T \cup \{i\}}$, then by symmetry, with probability $1/(k+1)$, $i$ may not be in the optimal selection of $k$ columns. Furthermore, since $T \cup \{i\}$ contains $k+1$ random columns, the expectation of the best rank-$k$ approximation to $A_{T \cup \{i\}}$ is at most a $(k+1)/n$ fraction of the optimal rank-$k$ approximation cost for $A$. By Markov's inequality, we obtain Equation (1).

- Lines 168 to 179: at a high level, in the analysis of [62], the authors need to conceptually normalize the columns to unit norm. However when the loss function is not scale-invariant, their algorithm is not able to normalize and thus the previous analysis completely breaks, as we explain starting at line 180. This is why we need to develop new techniques for analyzing our algorithm.

[Meta-Review · NeurIPS 2019]

The reviewers are broadly positive about the paper. However, the authors must take the following into account when preparing the final version: - Consider change of title: The reviewers felt that the "zero-one law" in the title was confusing and possibly even misleading - The reviewers are not convinced that the necessary and sufficient conditions are actually proved in the paper. These must be made very clear; or the claim of having necessary and sufficient conditions removed.